# Measuring sub-nanometer undulations at microsecond temporal resolution with metal- and graphene-induced energy transfer spectroscopy

Tao Chen [1], Narain Karedla [2,3] & Jörg Enderlein [1,4] ✉

Out-of-plane fluctuations, also known as stochastic displacements, of biological membranes play a crucial role in regulating many essential life processes within cells and organelles. Despite the availability of various methods for quantifying membrane dynamics, accurately quantifying complex membrane systems with rapid and tiny fluctuations, such as mitochondria, remains a challenge. In this work, we present a methodology that combines metal/graphene-induced energy transfer (MIET/GIET) with fluorescence correlation spectroscopy (FCS) to quantify out-of-plane fluctuations of membranes with simultaneous spatiotemporal resolution of approximately one nanometer and one microsecond. To validate the technique and spatiotemporal resolution, we measure bending undulations of model membranes. Furthermore, we demonstrate the versatility and applicability of MIET/GIET-FCS for studying diverse membrane systems, including the widely studied fluctuating membrane system of human red blood cells, as well as two unexplored membrane systems with tiny fluctuations, a pore-spanning membrane, and mitochondrial inner/outer membranes.

Cellular membranes play a crucial role in many fundamental physiological processes, such as cell adhesion, migration, signaling, and trafficking. From a physics perspective, membranes are selectively permeable, viscoelastic interfaces that surround cells and cellular organelles. They can be thought of as two-dimensional fluids composed mostly of lipids and membrane proteins that undergo Brownian motion within the plane of the membrane. Despite their complex structures, cellular membranes behave as if they are nearly perfect elastic shells, exhibiting linear viscoelastic responses and thermally induced spatial fluctuations, which are primarily bending fluctuations known as undulations.

Researchers have only shown in the past three decades that the non-equilibrium nature of membrane fluctuations can be attributed to active sources. These sources include cytoskeletal dynamics, protein and ion pumps, lipid transport systems, ATP-driven membrane remodeling, and the membrane-fission and fusion of cargo vesicles[1-8]. Brochard and Lennon presented the first quantitative description of the flickering dynamics of red blood cell membranes[9]. Since then, this description has been further extended in numerous works to include active force generation from point sources such as ion channels, transporters, and other factors[10-13]. However, these fluctuations have a relatively wide temporal range, and current theoretical models lack the

[1]Third Institute of Physics – Biophysics, Georg August University, Friedrich-Hund-Platz 1, Göttingen 37077, Germany. [2]The Rosalind Franklin Institute, Harwell Campus, Didcot OX11 OFA, UK. [3]Kennedy Institute of Rheumatology, University of Oxford, Roosevelt Drive, Oxford OX3 7LF, UK. [4]Cluster of Excellence "Multiscale Bioimaging: from Molecular Machines to Networks of Excitable Cells" (MBExC), Universitätsmedizin Göttingen, Robert-Koch-Str. 40, Göttingen 37075, Germany. ✉e-mail: jenderl@gwdg.de

necessary complexity to describe them quantitatively (see the review by Turlier et al.[14]).

Measuring cell membrane fluctuations provides valuable insights into various aspects of cellular health, including mechanical properties, disease states, drug responses, and fundamental biological processes[15]. By comprehending these fluctuations, one can evaluate cellular function, detect biomarkers for diseases, optimize drug development and screening, and enhance our understanding of basic biological processes[16–20]. This knowledge will have far-reaching implications for fields such as cell biology, biophysics, biomedical research, and medicine.

Several methods have been used to measure membrane fluctuations[21]. Camera-based wide-field detection methods such as contour analysis[22], fluorescence interference contrast (FLIC) microscopy[15,23,24], flicker spectroscopy[9], diffraction phase microscopy (DPM)[3], and reflection interference contrast microscopy (RICM)[25–27] capture membrane dynamics with millisecond temporal resolution (normally around ~ 10 ms), and with approximately 10 nm resolution in the case of RICM. In contrast, single point detection methods such as time-resolved membrane fluctuation spectroscopy (TRMFS)[4,28] or dynamic optical displacement spectroscopy (DODS)[21] achieve temporal resolutions of ~ 1 – 10 μs, but do not provide the spatial information of camera-based methods. Atomic force microscopy (AFM) is another powerful technique for studying the mechanical properties of biological materials, including stiffness, viscoelasticity, hardness, and adhesion. However, AFM is usually not suitable for tracking fast membrane fluctuations[15].

Most experimental work so far has focused on measuring membrane fluctuations on deflated giant unilamellar vesicles (GUVs) or red blood cells under stringent physiological conditions that lead to large amplitude bending fluctuations between 50 nm to 100 nm due to vanishing membrane surface tension. However, membranes of adherent cells or organelles that are under tension show tiny fluctuations with an amplitude below 10 nm at frequencies above 10 kHz[29]. Nevertheless, as of now, there have been no reports uncovering these specific fluctuations.

In this study, we present a method for quantifying nanoscale and sub-nanoscale membrane fluctuations with sub-millisecond temporal resolution using metal-/graphene-induced energy transfer (MIET/GIET). MIET/GIET relies on the electrodynamic coupling of a fluorescent emitter to either surface plasmons in a thin metal film (MIET)[30,31] or excitons in a single sheet of graphene (GIET)[32,33], see Fig. 1c, d. This coupling leads to a strong modulation of the excited state lifetime of the fluorescent dye that depends on its distance from the metal or graphene layers. By measuring the excited state lifetime of the dye, we can determine its distance from the metal or graphene layers, and convert this lifetime into a distance using the well-understood physics of electrodynamics. This method is a significant improvement over previous techniques and provides insight into the dynamics of cellular membranes.

In previous studies, MIET has been primarily used for structural analysis, such as mapping the topography of the basal membrane of living cells[31], reconstructing focal adhesions and stress fibers in three dimensions[34], measuring the distance between the inner and outer envelope of the nucleus[35], visualizing the dynamics of epithelial-mesenchymal transitions (EMT)[36], and enabling single-molecule localization and co-localization along the optical axis[37,38]. Additionally, MIET has been utilized to map the basal membrane and lamellipodia of human aortic endothelial cells[39,40] and to achieve three-dimensional isotropic resolution imaging of microtubules and clathrin pits in combination with SMLM[41]. By replacing the thin metal film with a single sheet of graphene (Graphene-Induced Energy Transfer or GIET), the localization accuracy can be improved by an order of magnitude within ~25 nm from the substrate[32]. GIET has been employed to measure the thickness of supported lipid bilayers with different lipid

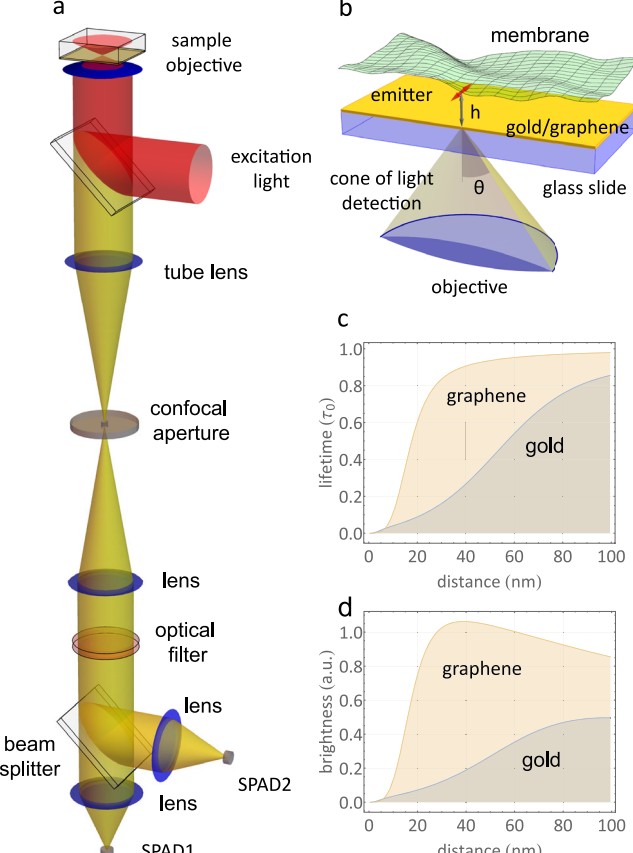

**Fig. 1 | Principle of MIET/GIET. a** Schematic of the MIET/GIET measurement setup. Fluorescence excitation and detection are performed through the coverslip from below, using a high numerical aperture objective. A conventional confocal fluorescence microscope equipped with a multichannel picosecond event timer for measuring fluorescence lifetimes is used for the fluorescence intensity and lifetime measurements. SPAD: single-photon avalanche diode. **b** Schematic of a fluctuating membrane above a gold film. The red arrow represents an emitter dipole on the membrane. **c** Calculated relative fluorescence lifetime $\tau/\tau_0$ of an emitter as a function of its distance $h$ from the surface of a 10 nm gold film or single sheet of graphene on glass. **d** Calculated relative brightness of an emitter as a function of its distance $h$ from the surface of a 10 nm gold film or single sheet of graphene on glass. All calculations were performed for an emitter with maximum emission wavelength of 680 nm, fluorescence quantum yield of 0.8, and random orientation. Source data for **c** and **d** are provided as a Source Data file.

compositions, enabling an accuracy of a few ångstroms[33], as well as to observe small changes in mitochondrial membrane organization upon respiration[42], or to quantify cholesterol-induced thickness changes of lipid bilayers[43].

To measure rapid membrane fluctuations, we combine MIET/GIET with fluorescence correlation spectroscopy (FCS)[44], using the setup shown in Fig. 1a. Our method is based on the fact that MIET/GIET not only modulates the fluorescence lifetime of dyes close to the substrate (Fig. 1c), but also the brightness (Fig. 1d), leading to strong fluctuations in fluorescence intensity when a fluorescently labeled membrane close to a MIET or GIET substrate fluctuates perpendicular to the substrate's surface, see Fig. 1b. FCS allows for the precise measurement and quantification of these fluctuations, which appear as a partial correlation decay on the time-scale of the membrane fluctuations. This approach is particularly useful for studying rapid membrane dynamics with sub-millisecond temporal resolution and nanoscale and sub-nanoscale spatial accuracy.

We demonstrate the method and validate its spatial and temporal resolution by measuring membrane fluctuations of model giant

unilamellar vesicles (GUVs). We then apply MIET/GIET-FCS to three different systems: (i) investigating the flickering of red blood cell (RBC) membranes with and without ATP; (ii) studying the fluctuations of pore-spanning membranes with small fluctuation amplitudes at high frequency; and (iii) exploring the undulations of the outer and inner membranes in active and resting-state mitochondria.

## Results

### Theory of MIET/GIET and height correlation function

Consider an electric dipole emitter that fluoresces (emits light) and is positioned at a distance $h$ above the surface of a MIET/GIET substrate, as depicted in Fig. 1b. When the distance $h$ decreases to a certain value (below 200 nm for a gold film, or below 25 nm for graphene), the electric near field of the emitter can stimulate surface plasmons in the metal film or excitons in graphene, causing the emitter to de-excite more quickly from its excited state to the ground state. This phenomenon can be studied by solving Maxwell's equations for the electromagnetic field of the electric dipole emitter in the presence of the MIET/GIET substrate, as outlined in refs. 33,37. This computation yields several important quantities.

Firstly, the total emission power $S(h, \beta)$ of the emitter can be determined as a function of its distance $h$ from the substrate and its orientation, described by the angle $\beta$ between its dipole axis and the vertical direction. This is achieved by integrating the (time-averaged) Poynting vector, which is the cross product of the electric and magnetic field vectors, multiplied by the speed of light and divided by $8\pi$, across two horizontal planes within the emitter's medium (buffer solution) that enclose the emitter from both sides.

Secondly, the total far-field emission $N(h, \beta)$ that is radiated into the lower half space (glass coverslip) can be calculated by integrating the Poynting vector over the interface between the glass and the metal/graphene layer.

Lastly, the relative proportion $C(h, \beta)$ of the emission that is collected by the objective lens, which is the part that is radiated into its cone of light detection, can be determined by integrating the angular distribution of emission (the Poynting vector as a function of the emission angle) over this cone of light detection.

This emission power $S(h, \beta)$ is proportional to the radiative transition rate of a fluorescent molecule from its excited to its ground state. By knowing the quantum yield of fluorescence $\phi$ (ratio of radiative to total transition rate), one can calculate the observable fluorescence lifetime $\tau_f$ as follows:

$$\frac{\tau_f(h,\beta)}{\tau_0} = \frac{S_0}{\phi S(h,\beta) + (1-\phi)S_0}. \tag{1}$$

In this equation, $S_0 = cnk_0^4 p^2/3$ represents the emission power of an electric dipole emitter with unity quantum yield in a homogeneous unbounded dielectric medium with refractive index $n$ (which is equal to the refractive index of the medium above the MIET substrate), where $c$ is the speed of light, $k_0$ is the wave vector in vacuum, and $p$ is the amplitude of the dipole moment vector. $\tau_0$ is the corresponding free-space fluorescence lifetime. The angular dependence of $S(h, \beta)$ can be decomposed as $S(h,\beta) = S_\perp(h)\cos^2\beta + S_\parallel(h)\sin^2\beta$, where $S_\perp(h)$ and $S_\parallel(h)$ are the radiative emission rates of emitters oriented perpendicular and parallel to the substrate, respectively, and are now functions of $h$ only. When a molecule is freely rotating (i.e., its reorientation rate is much faster than its fluorescence lifetime), the full emission rate can be simplified to $S(h) = [S_\perp(h) + 2S_\parallel(h)]/3$ after averaging over all possible orientations (see Supplementary Fig. 8).

The curves in Fig. 1c demonstrate the calculation of the relative lifetime $\tau_f(h)/\tau_0$ for a rapidly rotating dye with a quantum yield of $\phi = 0.8$ emitting at $\lambda_{em} = 680$ nm as a function of height $h$ over a gold-coated and a graphene substrate. As can be observed, the fluorescence lifetime increases as a monotonic function within a distance of up to approximately 150 nm for the gold layer and up to 25 nm for graphene.

In nearly all applications of MIET and GIET thus far, the dependency of lifetime on distance has been utilized to localize an emitter along the optical axis by measuring its lifetime. However, the near-field coupling and energy transfer of an emitter to a metal or graphene layer not only modulates its excited-state lifetime but also affects its observable brightness. By determining the total far-field emission $N(h, \beta)$ into the glass and its relative portion $C(h, \beta)$ collected by the objective, one can calculate the brightness value $b(h)$ of a randomly oriented emitter using the following equation (up to some proportionality constant):

$$b(h) \propto \int_0^{\pi/2} d\beta \sin\beta \frac{N(h,\beta)}{S(h,\beta)} C(h,\beta) \tag{2}$$

Similar to the case of $S(h, \beta)$, the function $N(h, \beta)$ can be decomposed as $N_\perp(h,\beta)\cos^2\beta + N_\parallel(h,\beta)\sin^2\beta$, and similarly for $C(h, \beta)$. A comprehensive description of all the mathematical details of this calculation is presented in ref. 45.

The graph in Fig. 1d illustrates the relative brightness $b(h)$ of a dipole under the same conditions as the lifetime calculations, but with an objective having an numerical aperture (N.A.) of 1.49 and refractive index $n_{imm}$ of immersion oil of 1.52. It should be noted that when the dipole is far from the metal or graphene surface and the energy transfer is negligible, the relative brightness approaches the net transparency of the substrate. We verified the lifetime and relative brightness calculations using additional experiments where we measured these parameters on bilayers for various thickness of silica spacers (see Supplementary Fig. 1 and Supplementary Note 1).

The distance-dependent detectable fluorescence brightness is of particular significance in the current study, as it enables the combination with FCS to measure minute vertical membrane fluctuations near a MIET/GIET substrate in a fast and precise manner. In the absence of a MIET/GIET substrate, FCS would primarily capture intensity fluctuations resulting from the lateral diffusion of labeled lipids through the confocal volume. However, in the presence of a MIET/GIET substrate, these intensity fluctuations are also influenced by vertical position fluctuations of the emitters. Furthermore, high label densities of fluorescently labeled lipid membranes render the impact of lateral molecular diffusion on the observable fluorescence intensity completely negligible, and the resulting auto-correlation function (ACF) measured in FCS experiments is dominated by the vertical position fluctuations of the membrane.

The intensity correlation function $g_I(t)$ can be expressed as the time average of the product of intensity fluctuations $\delta I(t') = I(t') - \langle I \rangle$ and $\delta I(t' + t) = I(t' + t) - \langle I \rangle$, denoted by $\langle \delta I(t')\delta I(t' + t)\rangle_{t'}$, where the angular brackets indicate time averaging. When the time-dependent height of the membrane, $h(t) = h_0 + \delta h(t)$, falls within a range where the intensity-versus-height relationship can be approximated by a linear function, the height correlation function $g_h(t) = \langle \delta h(t')\delta h(t' + t)\rangle_{t'}$ can be related to $g_I(t)$ through the approximation

$$g_I(t) \propto \left(\frac{dI}{dh}\right)^2_{h=h_0} g_h(t). \tag{3}$$

Thus, measuring the intensity correlation function $g_I$ does, in the linear regime, allow for determining the height correlation function $g_h$. The average height $h_0$ itself can be estimated from the average fluorescence lifetime using the lifetime-versus-height MIET/GIET relationship, see Fig. 1c. As can be seen from Fig. 1d, the distance range where the r.h.s. of Eq. (3) is an acceptable approximation is ca. 30–70–nm for MIET (gold layer), and ca. 10–25 nm for GIET (graphene). We also used GUV membrane measurements for experimentally checking the

validity of assuming a linear relationship between height and intensity, see Supplementary Fig. 2.

The ordinate of the height correlation function at lag time $t = 0$ yields the root mean square amplitude of displacement $\psi = \sqrt{\langle \delta h^2 \rangle}$ and the relaxation time $\tau^*$ which is determined by the point where this function has decreased to one-half of its maximum value.

In the next two subsections, we will use MIET for measuring membrane fluctuations of GUVs and cells. In all these experiment, the used MIET substrate has always the same structure: On a glass cover slide (refractive index 1.52), we deposited a 10 nm thick gold film which was covered by a 10 nm thick protecting quartz layer.

## MIET-FCS: membrane fluctuations in GUVs

We assessed the performance of MIET-FCS by measuring membrane fluctuations of GUVs labeled with fluorescent markers[21]. A typical geometry of the experiment is shown in Fig. 2a. We first showed that a densely labeled membrane leads to a flat ACF with no lag-time dependence. For this purpose, we prepared GUVs from SOPC and fluorescently labeled DPPE (DPPE-Atto655) by electroformation[33] in a sucrose solution of 230 mOsm/L. We used two different concentrations of fluorescently labeled lipids (DPPE-Atto655): $10^{-3}$ mol% and 1 mol%. We immersed GUVs into a measurement chamber containing an isotonic solution, which had an osmolarity equivalent to that of the GUVs. Most of the GUVs settle down at the bottom of the chambers within 15 min forming a flattened contact area with the surface (see Fig. 2b, c). FCS measurements were recorded at the bottom of the chamber (glass cover slide covered with BSA) with the excitation focus of the confocal microscope placed within the contact area (glass surface). Figure 2d shows a comparison of the measured intensity correlation functions for both label concentrations. At $10^{-3}$ mol% labeling concentration, we observed a temporally decaying correlation due to the diffusion of labeled lipids in and out of the focus, with a diffusion coefficient of $7.3 \pm 0.3 \, \mu m^2 s^{-1}$ (mean ± standard deviation), which is close to reported literature values[46]. As the amplitude of the diffusion-related part of the correlation function scales with the inverse number of molecules in the focal volume[47], the diffusion-related decay of the correlation function is suppressed at high label concentration (1 mol%).

Next, we performed proof-of-principle experiments to show that MIET-FCS is indeed capable of significantly amplifying the amplitude of minute membrane fluctuations that cannot be observed by FCS on glass substrates with GUVs, and are barely visible when positioning the focus slight below the glass-water interface as in a DODS experiment. In DODS, the membrane is positioned in such a way that its mean position is located at the inflection point of the excitation intensity profile. This leads to measurable intensity fluctuations when the membrane undergoes strong bending fluctuations. In particular, we immersed GUVs (230 mOsm/L) into a hypertonic solution with an osmolarity of 400 mOsm/L, which deflates the GUVs and leads to strong membrane fluctuations. The deflation is clearly visible in the non-circular geometry of the contact area of a GUV settled on a BSA-covered glass surface, shown in Fig. 2b. From the xz scanning image, it was found the GUV spread take on a spherical shape (see Supplementary Fig. 11). In Fig. 2e we show the comparison of three FCS measurements with the following arrangements: GUV on glass with the excitation focus located at the substrate surface; GUV on glass with the excitation focus positioned much below the glass surface (as used in DODS experiments); GUV on a MIET substrate with the excitation focus on the substrate surface. As anticipated, we did not observe any lag-time-dependent correlation decay for the GUV on glass when the membrane was in focus, and we observed a very small decay amplitude ($<10^{-3}$) for the GUV on glass but with defocused excitation. In contrast, the FCS measurement on a MIET substrate yields a correlation function

with considerable amplitude (-0.02), that is an order of magnitude larger. Moreover, from the DODS measurement, it is difficult to derive a precise number for the root mean square height fluctuation since this value is sensitive on the exact vertical position of the laser focus. In contrast, using the mean lifetime values of the GUV measured on the MIET substrate, we can calculate a mean height value, which in this example is about $h_0$ - 37.0 nm. This allows us to use Eq. (3) to determine the value of the root mean square height fluctuation $\psi$ of 6.0 nm from the amplitude of the height correlation curve.

Next, we used MIET FCS to compare the membrane fluctuations of tense GUVs (immersed in an isotonic buffer with 230 mOsm/L) and deflated GUVs (immersed into a buffer with of 400 mOsm/L). Figure 2f shows three TCSPC measurements of the fluorescence decays for a tense GUV on pure glass, on a MIET substrate, and for a deflated GUV on a MIET substrate. One can clearly see the MIET induced reduction of the fluorescence lifetime as compared to the pure-glass measurement for both, tense as well as deflated GUVs. Moreover, a tense GUV on a MIET substrate shows a much shorter fluorescence lifetime as compared to a deflated GUV on the same substrate (0.49 ns for tense GUV and 1.43 ns for deflated GUV). We converted these fluorescence lifetimes into height values, which yields $h_0 = 2.3 \pm 0.3$ nm ($N = 23$) for tense GUVs, and $h_0 = 36.1 \pm 3.3$ nm ($N = 36$) for deflated GUVs from BSA layer surface, which signifies that the average height of the basal membrane of the tense GUVs is much smaller than that of the deflated GUVs (see inset of Fig. 2f). Figure 2g shows a comparison of the height correlation curves $g_h$ for the tense and deflated GUVs on the MIET substrate. As can be seen, the height fluctuations for the tense GUV are nearly negligible ($0.39 \pm 0.11$ nm, $N = 23$), while the deflated GUV shows by orders of magnitude larger fluctuations with a root mean square height fluctuation of $\psi = 8.0 \pm 2.6$ nm ($N = 36$), see also inset of Fig. 2g. This is in excellent agreement with published values for SOPC obtained using RCIM ($h_0 = 31$–$39$ nm and $\psi = 4$–$15$ nm)[48]. Interestingly, our SOPC results are very similar to published results on DMPC using FLIC microscopy ($h_0 = 30$–$60$ nm and $\psi = 10$ nm)[15,23,24].

The relaxation time $\tau^*$, defined as the lag time at which the height correlation curve has decreased to half its maximum value, was determined to be $50 \pm 30$ ms using MIET FCS for the deflated membrane, which is consistent with previous findings in the literature[21].

We fitted the measured height correlation function $g_h(t)$ with a theoretical model described in refs. 48–51. The fluctuations are determined by the membrane bending rigidity $\kappa$, the membrane tension $\sigma$, the effective viscosity $\eta$, and an effective interaction potential $\gamma$ describing the interaction between the fluctuating membrane and the substrate surface. Then, the height correlation function is described by the following equation:

$$\langle \delta h(t') \delta h(t' + t) \rangle_{t'} = \frac{k_B T}{2\pi} \int_{q_{min}}^{q_{max}} dq \frac{q}{\kappa q^4 + \sigma q^2 + \gamma} \exp\left[ -\Gamma(q)t - \frac{1}{4} w^2 q^2 \right] \tag{4}$$

where $w$ denotes the diameter of the confocal detection area. The function $\Gamma(q)$ is defined by

$$\Gamma(q) = \frac{\kappa q^4 + \sigma q^2 + \gamma}{2\eta q} \frac{\sinh^2(q h_0) - (q h_0)^2}{\sinh^2(q h_0) + \sinh(2 q h_0)/2 + q h_0 (1 - q h_0)} \tag{5}$$

The integration boundaries of the integral in Eq. (4) are $q_{min} = (\gamma/\kappa)^{1/4}$ and $q_{max} = 1/h_0$[51].

For SOPC membranes, the value of the bending modulus $\kappa = 20 \, k_B T$ is well known and was kept constant throughout our analysis[21,52]. The effective viscosity $\eta = 1.2$ mPas was calculated as the arithmetic mean of the viscosity of the buffer solutions inside and outside the GUV. The diameter of the confocal detection area in our measurement was determined as $w = 280$ nm. We measured and fitted $N = 36$ height fluctuation correlation curves, a typical curve and fit is

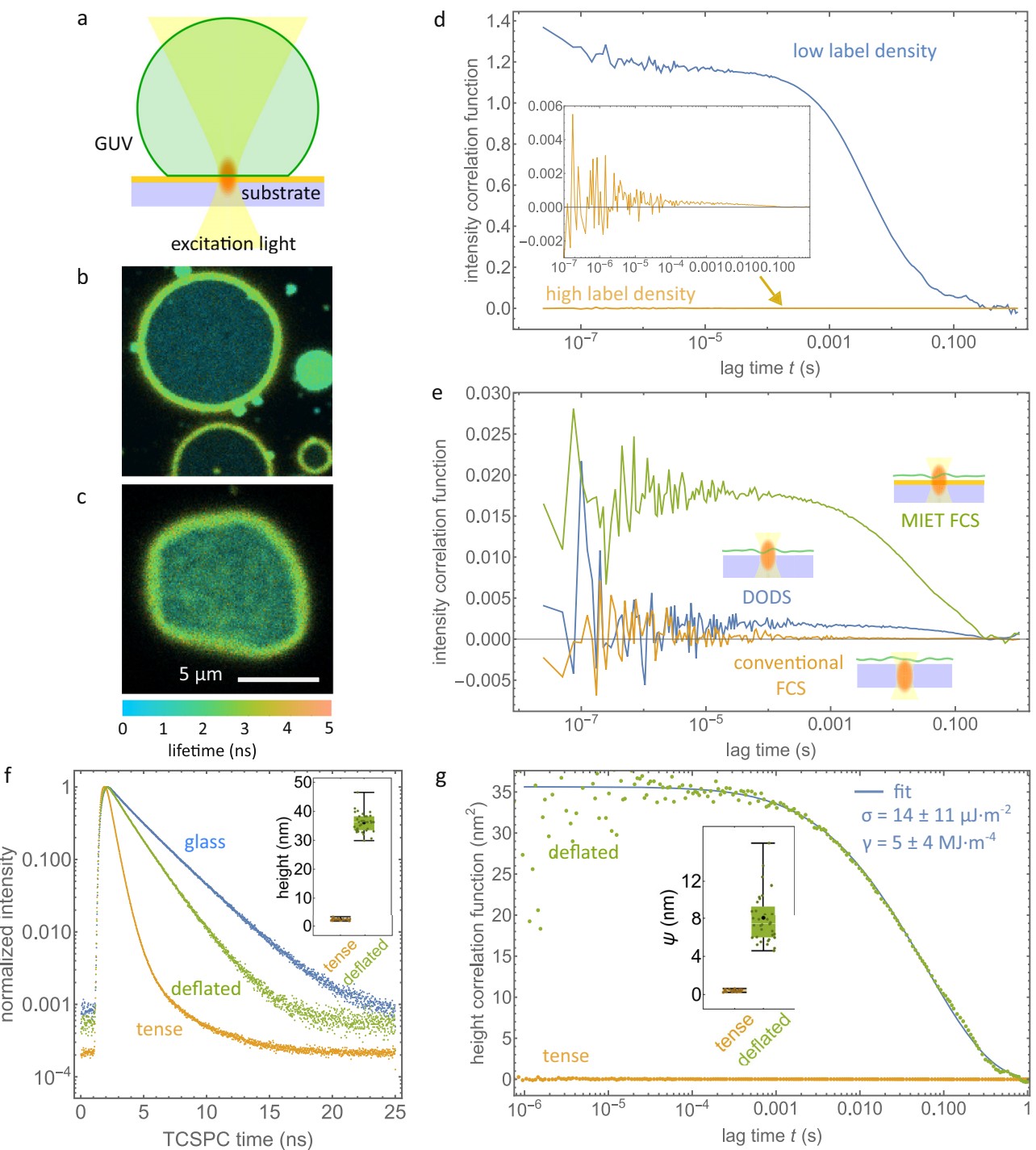

**Fig. 2 | MIET-FCS measurement on model GUV membrane. a** Schematic of a MIET-FCS experiment on a GUV attached to the MIET substrate. **b** Fluorescence lifetime image of a tense GUV showing a ring of increased intensity where the GUV membrane first touches the surface. Fluorescence lifetime is shown by color, intensity by brightness. **c** Same as **b**, but for a deflated GUV. **d** Intensity correlation curves measured on the proximal membrane of GUVs immobilized on a pure glass substrate. Shown are two curves for membranes with low and high label density. Inset shows the enlarged plot of the membrane with high label density. **e** Comparison of correlation curves measured on the proximal membrane of surface-immobilized GUVs obtained on pure glass substrate with the excitation focus on the glass surface (conventional FCS) or ca. 1 µm below (DODS), and obtained on a MIET substrate. **f** Fluorescence lifetime decay curves for tense GUVs

on glass and MIET substrate, and for a deflated GUV on MIET substrate. Inset presents corresponding heights ($h_0$). **g** Measured MIET-FCS correlation curves (dotted lines) for tense and deflated GUVs. Inset presents corresponding height fluctuation amplitudes ($\psi$). Fit values from a fit (solid line) of Eq. (4) to the deflated GUV curve are also shown. Data are presented as mean values ± SD. The experiment in **b** was repeated two times with similar results and the experiment in **c** was repeated four times with similar results. In **f** and **g**, box plots show the 25th–75th quantiles (box), median (white line), mean (black dot), and whiskers (minima to maxima). $n = 36$ independent measurements over 4 independent experiments for deflated GUV and $n = 23$ independent measurements over 2 independent experiments for tense GUV. Source data for **f**, **d**, **e**, and **g** are provided as a Source Data file.

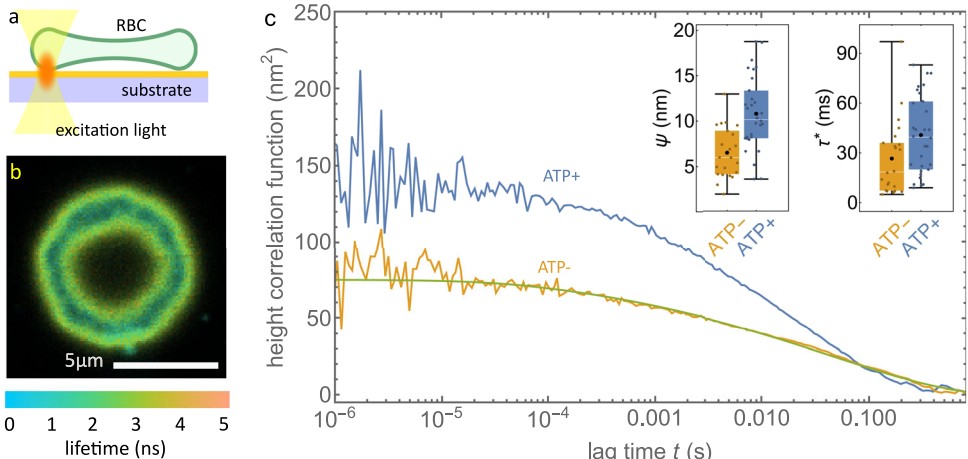

**Fig. 3 | MIET-FCS measurement on RBC membrane. a** Schematic of the MIET-FCS experiment of membrane fluctuations in a living RBC. An RBC has settled on the BSA covered surface of a MIET substrate. **b** Fluorescence lifetime image of a RBC on a MIET substrate. Lifetime is shown by color, fluorescence intensity by brightness. The cell's rim touching the surface is visible as a ring of enhanced fluorescence intensity and reduced lifetime. **c** Height fluctuation correlation curves measured on the membrane of an ATP-saturated and an ATP-depleted RBC. Fit curve from a fit (solid line) of Eq. (4) to the ATP-depleted RBCs. The experiment in **b** was repeated four times with similar results. Insets show box plots of fluctuation amplitudes ($\psi$) and relaxation times ($\tau^*$). Box plots show the 25th–75th quantiles (box), median (white line), mean (black dot), and whiskers (minima to maxima). $n = 30$ independent measurements over 4 independent experiments for ATP+ and $n = 22$ independent measurements over 3 independent experiments for ATP-. Source data for **c** is provided as a Source Data file.

shown in Fig. 2g. From these fits, we found a surface tension value of $\sigma = 14 \pm 11 \, \mu J m^{-2}$ and an interaction potential strength value of $\gamma = 5 \pm 4 \, MJm^{-4}$, in excellent agreement with published results for similar GUV measurements[48–50].

## MIET-FCS: membrane fluctuations of red blood cells

We applied MIET FCS to a well-studied biological system by measuring membrane fluctuations in red blood cells (RBCs). It is well-known that RBC membranes exhibit significant bending fluctuations, which have been extensively studied in the past using various techniques. These include studies conducted by Park et al.[3] and Betz et al.[4], as well as Monzel et al.[21,53] and Gov et al.[53].

When RBCs settle on a planar surface, the contact zone between their membrane and the surface forms a circular rim-like structure, owing to the concave shape of these cells. This can be seen in Fig. 3a, b. A high label density is used here again to ensure that the lateral diffusion did not affect our measurement (see Supplementary Fig. 3).

We conducted MIET-FCS measurements on the proximal membrane of the circular rim formed by RBCs settled on a BSA-coated MIET substrate. This enabled us to measure the membrane fluctuations in the presence and absence of ATP, as shown in Fig. 3c. The average height values of the membrane did not appear to be affected by the RBC's biological activity, with values of $36.4 \pm 5.5$ nm in the absence of ATP and $37.7 \pm 4.7$ nm in the presence of ATP.

However, we observed that membrane fluctuation amplitudes were larger in the presence of ATP, with an increase of 1.6 times as compared to ATP-depleted RBCs. Specifically, the amplitude values were $6.5 \pm 2.8$ nm and $10.8 \pm 4.1$ nm in the absence and presence of ATP, respectively. The measured fluctuation amplitudes for both ATP-saturated and ATP-depleted cells as obtained with MIET-FCS are significantly smaller than those obtained using other methods, such as DODS (74 nm for ATP+ and 41 nm for ATP-)[21] or DPM (48 nm for ATP+ and 32 nm for ATP-)[3]. The reason for these smaller amplitudes in our measurements is due to the fact that we can measure only the basal membrane of the RBCs, which is influenced by the presence of the substrate that dampens these fluctuations[54]. Additionally, the relaxation time was measured to be $\tau^* = 27 \pm 23$ ms for ATP-depleted cells, which was slightly smaller than the value of $41 \pm 28$ ms in the presence of ATP. This suggests a faster dissipation of thermal-driven fluctuations for active cells. This observation is consistent with published

DODS[21] and optical tweezer measurements[17]. It is assumed that active ATP-powered processes result in a slower-than-thermal fluctuation dynamics, which could be e.g. the active motion of the spectrin network, as discussed in ref. 17.

We used Eq. (4) to fit the height correlation functions for ATP-depleted RBCs, see also Supplementary Note 4, yielding values for membrane tension ($2.7 \pm 2.1 \, \mu J/m^2$) and bending modulus ($(3.9 \pm 2.7) \times 10^{-20}$ J). Our results are in fair agreement with published values using other methods/techniques, see Supplementary Table 1. We did not fit data from ATP-saturated RBCs because equilibrium theory is not suitable to model active ATP-driven membrane fluctuations[17].

The previous two subsections employed MIET, a technique based on the metal-induced quenching of fluorescence, to measure membrane fluctuations in GUVs and red blood cells. In the following two subsections, we will use GIET, which relies on the energy transfer from excited fluorescent dyes to excitons in a single sheet of graphene.

The dynamic range of GIET is approximately 8 times smaller than that of MIET, at around 20 nm compared to 160 nm. However, this limitation allows GIET to achieve greater sensitivity to vertical motions of an emitter, making it ideally suited for measuring small fluctuations down to the Ångström length scale.

## GIET-FCS: fluctuations of a pore spanning membrane

SLBs are a gold standard for in vitro studies of lipid membranes. However, the direct contact between the membrane and the solid support can significantly alter the dynamics and properties of SLBs. Therefore, it is important to be able to study free-standing membranes that are not disturbed by solid supports. One way for achieving this is using pore-spanning membranes (PSMs), where a membrane is formed over small pores in a solid substrate, as shown in Fig. 4a.

Compared to GUVs, PSMs typically have a much larger surface tension, which reduces their transversal fluctuations by orders of magnitude[55,56]. Currently, no technique has been reported that can measure such small fluctuations in PSMs.

We utilized pore-spanning membranes (PSMs) to prepare free-standing membranes away from a solid support, which can be important when studying membrane properties that may be affected by contact with a solid support. To create PSMs, we deposited negatively charged giant unilamellar vesicles (GUVs) onto a positively

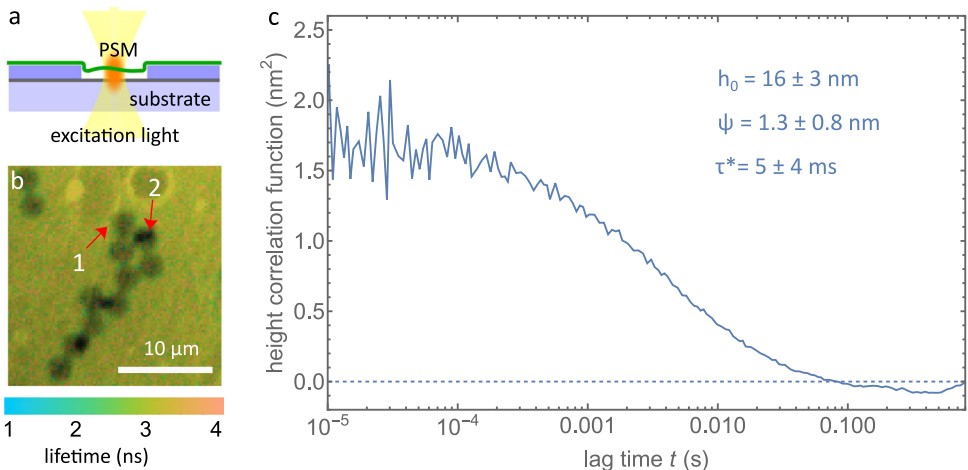

**Fig. 4 | GIET-FCS measurement on PSM. a** Schematic of a GIET-FCS experiment on a PSM. The GIET substrate consists of a single sheet of graphene sandwiched between a glass cover slide and a 32 nm thin SiO₂ spacer layer. Into that SiO₂ layer, pores were etched with a depth of 30 nm and a diameter of 3 μm. **b** Fluorescence lifetime image of a PSM sample. Lifetime is shown by color, fluorescence intensity by brightness. Several pores are visible as dark disks on a brighter background. Two types of pores with different intensity/lifetime can be distinguished, corresponding to free-standing membranes (labeled with "1" in panel **b**) and membranes touching the pore bottom (reduced intensity/lifetime, labeled with "2" in panel **b**). **c** Height fluctuation correlation curves for a PSM on a GIET substrate. Data in **c** are presented as mean values ± SD. The experiment in **b** was repeated two times with similar results. Source data for **c** is provided as a Source Data file.

charged substrate with pores (see Materials and Methods)[57]. This substrate consisted of a single sheet of graphene sandwiched between a glass cover slide and a thin SiO₂ spacer layer with a thickness of 32 nm, into which pores of 3 μm diameter and 30 nm depth were prepared via the nanosphere lithography method[58], as shown in Fig. 4b. After deposition, the GUVs ruptured to form PSMs.

We confirmed the free-standing nature of the PSMs using GIET and GIET-FCS by measuring their average height and fluctuation amplitudes (see Supplementary Fig. 5 and Supplementary Note 2). The fluorescence lifetime image in Fig. 4b clearly distinguishes pores showing a short and long lifetime, representing membranes touching the pore bottom and free-standing membranes, respectively. From lifetime measurements, we found an average height of $16.3 \pm 2.6$ nm ($N = 20$) for the free-standing membranes, indicating that the membrane bends into the pore due to membrane pre-stress as induced by the reduced hydrophilicity of the SiO₂ substrate[56,59–61]. We performed GIET-FCS measurements at the center of the PSMs, as shown by the solid line in Fig. 4c. As a control, we performed similar measurements on PSMs over pores without a graphene sheet (see Supplementary Fig. 5a). As expected, the control measurement did not show any intensity correlation amplitude or decay, in stark contrast to the GIET-FCS measurements (see Supplementary Fig. 6). In the latter measurements, we observe a height correlation amplitude of $1.3 \pm 0.8$ nm ($N = 20$) and a correlation relaxation time of $\tau^* = 4.5 \pm 3.7$ ms. However, the height fluctuation correlation curve of PSMs cannot be well fitted with the theoretical model used for GUV fluctuations, due to the more complex hydrodynamic interactions of the fluctuating membranes with the rim of the pore as well as the liquid enclosed in the membrane-covered pore.

We did not detect any significant fluctuations using GIET-FCS when the pore depth was reduced to 20 nm (see Supplementary Fig. 6). In this case, the PSMs are too close to the pore bottom, with an average height of $8.2 \pm 1.3$ nm, as determined from lifetime measurements. This proximity to the substrate dramatically reduces the amplitude of membrane fluctuations.

### GIET-FCS: fluctuations of mitochondrial membranes

Mitochondria are dynamic organelles that play a vital role in cellular energy production. They are characterized by a double-membrane structure[62]. The smooth outer mitochondrial membrane (OMM) surrounds the organelle and contains numerous protein-based pores, while the inner mitochondrial membrane (IMM) is deeply convoluted and contains the machinery responsible for ATP synthesis[62]. It has been shown that isolated mitochondria can maintain their biological activity, such as their ability to synthesize ATP, in the presence of suitable precursors in the surrounding buffer[63].

To investigate the effect of ATP-synthesis precursor molecules on mitochondrial membrane dynamics, we performed GIET-FCS experiments on isolated mitochondria in two different states: an "active" state with a buffer containing ADP molecules that promote ATP synthesis, and a "resting" state with a buffer lacking these molecules (see subsection "Mitochondria membranes measurement" in Methods). To label specific regions of the mitochondria, we attached fluorescent probes to either the outer mitochondrial membrane (OMM) or the inner mitochondrial membrane (IMM), as shown in Fig. 5a, b. To ensure that lateral diffusion did not affect our measurements, we used a label density that was sufficiently high (see Supplementary Fig. 7 and Supplementary Note 3).

We used a GIET substrate consisting of a single layer of graphene on a glass cover slide, with a 10 nm thick SiO₂ spacer layer for OMM measurements and a 2 nm thick SiO₂ spacer layer for IMM measurements. The use of a thinner SiO₂ spacer for IMM ensures that the IMM is positioned at the working distance of GIET. We conducted comparative GIET-FCS experiments on isolated mitochondria using a buffer containing ATP-synthesis precursor molecules (ADP+ active state) and a buffer without these molecules (ADP− resting state). We measured $N = 22$ mitochondria under ADP− and $N = 21$ mitochondria under ADP+ conditions for OMM, and $N = 21$ under ADP− and $N = 26$ under ADP+ conditions for IMM. Exemplary GIET-FCS curves are shown in Fig. 5c, and results for the average distance, height fluctuation amplitudes, and correlation relaxation times are presented in Fig. 5d–f.

Our results showed that the average membrane height $h_0$ for OMM remained almost constant for mitochondria in the resting ($7.8 \pm 1.7$ nm) and active ($8.0 \pm 1.9$ nm) states. However, this value decreased by about 1.3 nm for IMM ($16.5 \pm 0.7$ nm for ADP+ active and $17.8 \pm 1.6$ nm for ADP− resting states), consistent with previous reports[42]. The reduced distance between the OMM and IMM in the active state enhances transport and exchange of molecules between these two membranes. A similar observation has been reported using electron microscopic tomography[64]. It should be noted that when performing a point measurement with our confocal microscope, the

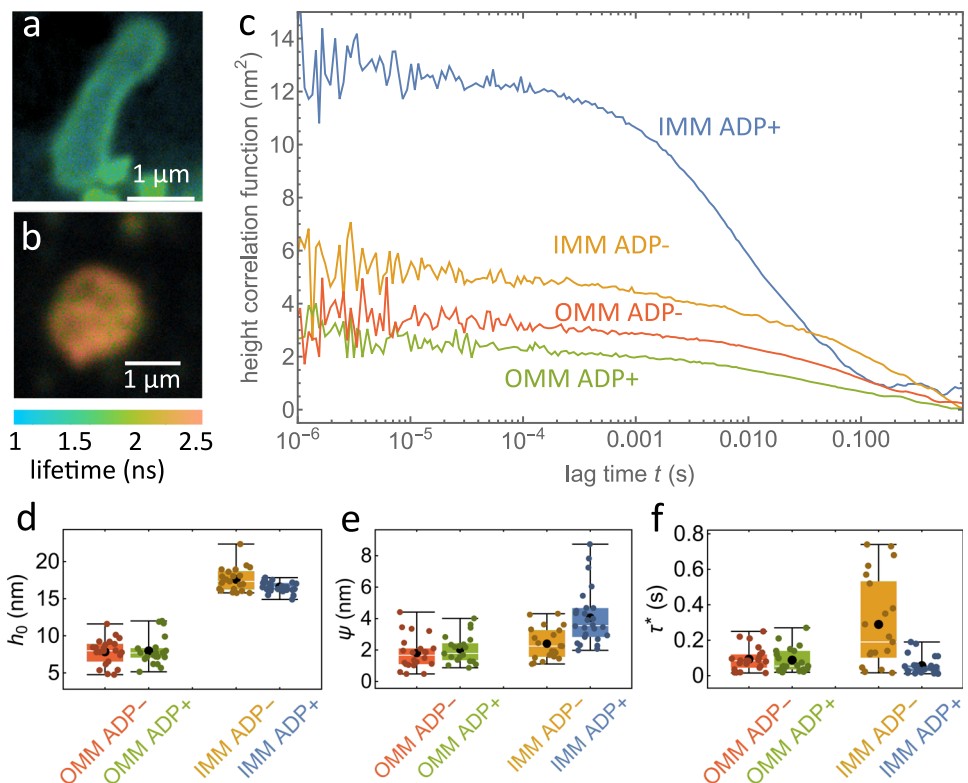

**Fig. 5 | GIET-FCS measurement on the mitochondrial membrane. a** FLIM image of the OMM above a graphene substrate. **b** Same as **a** for the IMM. **c** Height correlation curves for the IMM and OMM under ADP+ and ADP− conditions. **d** Average height values ($h_0$). **e** Height correlation amplitude, $\psi$. **f** Correlation relaxation times, $\tau^*$. The experiment in **a** was repeated two times with similar results and the experiment in **b** was repeated three times with similar results. In **d**–**f**, box plots show the 25th–75th quantiles (box), median (white line), mean (black dot), and whiskers (minima to maxima). $n = 22$ independent measurements over 2 independent experiments for OMM ADP-, $n = 21$ independent measurements over 2 independent experiments for OMM ADP+, $n = 21$ independent measurements over 3 independent experiments for IMM ADP-, $n = 26$ independent measurements over 3 independent experiments for IMM ADP+. Source data for **c**, **d**, **e** and **f** are provided as a Source Data file.

observed signal is a spatial average over the size of the excitation focus. Thus, for the membrane measurements on mitochondria, our results represent averages over lateral regions of ca. 300 nm, thus averaging out the highly convoluted geometry of the membranes. Using GIET-FCS, we found a similar trend for the height fluctuation amplitudes: OMMs exhibited comparable fluctuation amplitudes $\psi$ for mitochondria in their resting and active states ($1.8 \pm 1.0$ nm for ADP− resting and $2.0 \pm 0.9$ nm for ADP+ active states), while IMMs displayed enhanced fluctuation amplitudes in active state mitochondria ($2.4 \pm 1.0$ nm for ADP− resting and $4.1 \pm 1.7$ nm for ADP+ active states).

Finally, the correlation relaxation time ($\tau^*$) also exhibited a difference between the two membrane types. For IMMs, the values decreased from $0.29 \pm 0.25$ s in the resting state to $0.056 \pm 0.048$ s in the active state, while for OMMs, almost no change was observed ($0.088 \pm 0.072$ s for the ADP− resting and $0.093 \pm 0.066$ s for the ADP+ active states). These results suggest that the active membrane fluctuations of IMMs are enhanced in the mitochondria's active state, likely due to structural reorganization of the IMMs' cristae and mitochondrial volume regulation, as reported previously[42].

## Discussion

We have developed a method, MIET/GIET-FCS, which enables the study of lipid membrane fluctuations. This technique combines the nanometer spatial resolution of MIET/GIET with the microsecond temporal resolution of FCS. What sets our approach apart is its simplicity, as it can be implemented using a conventional fluorescence lifetime imaging confocal microscope. Additionally, our method eliminates the need for experimental calibration by providing exact MIET/GIET calibration curves based on sample geometry.

It should be emphasized that MIET/GIET measurements are based on fluorescence lifetime rather than intensity, and therefore, are much more robust with respect to labeling density and fluorescence intensity than purely intensity-based fluorescence methods. In our studies presented here, we suppressed lateral-diffusion related contributions to the correlation function by using rather large label concentrations. However, this is not absolutely necessary for applying MIET-GIET for measuring membrane fluctuations, it only simplifies final data analysis and improves signal to background ratio.

Through the application of MIET/GIET-FCS to various systems, including GUVs, living red blood cells, pore-spanning membranes, and the inner and outer membranes of living mitochondria, we have demonstrated its versatility for a wide range of membrane biophysical studies. With the high axial resolution of GIET, it could be even feasible to measure Ångström-size variations in nano-topography of substrate-supported membranes. By expanding MIET/GIET-FCS from a point measurement to fast scanning FCS or camera-based wide-field fluorescence lifetime image[65], we can achieve high-statistical power measurements of cell membrane fluctuations (see Supplementary Fig. 12). We hope that MIET/GIET-FCS will become a powerful and versatile tool for studying membrane biophysics, in particular nanoscale structure and microsecond fluctuations, in a broad range of contexts and applications.

## Methods

### Ethical statement

The research conducted in this study complies with all relevant ethical regulations. Human RBCs were voluntarily provided by a healthy donor (male, aged 34), who provided informed consent to have the

samples collected and used for research. Blood was collected and stored adhering to standard operation procedures, to minimize storage and sample handling artifacts. Biosampling was conducted according to the Declaration of Helsinki and according to the general instructions of the ethics committee of the University of Göttingen Faculty of Medicine, Göttingen, Germany. No ethics approval was judged necessary, since the procedure did not involve ethically-relevant issues.

## MIET and GIET measurements

All measurements, including lifetime measurements, were conducted using a self-constructed confocal setup. We used a pulsed diode laser (LDH-D-C 640, PicoQuant) with a pulse width of 50 ps FWHM and a repetition rate of 40 MHz, operating at an excitation wavelength of 640 nm. The excitation light was filtered using a clean-up filter (LD01-640/8 Semrock) and collimated through an infinity-corrected 4x objective (UPISApo 4X, Olympus). The laser beam was then directed to a high numerical aperture objective (UAPoN 100X, oil, 1.49 N.A., Olympus) via a dichroic mirror (Di01-R405/488/561/635, Semrock). The emission light was collected and focused into a 100 μm diameter pinhole and refocused onto two avalanche photodiodes (τ-SPAD, PicoQuant). We utilized a long-pass filter (BLP01-647R-25, Semrock) and two band-pass filters (Brightline HC692/40, Semrock) before the pinhole and the detector, respectively. The photo signals obtained from the detector were analyzed using a multi-channel picosecond event timer (Hydraharp 400, PicoQuant) with 16 ps time resolution.

In a typical measurement, we first scanned the sample at the coverslip surface to determine the contour of the GUV, cell, or mitochondrion and selected the appropriate point position for data acquisition. The laser power was adjusted to reach a maximum count rate of 50–250 kcps during data collection. For GUV, mitochondrion, and cell measurements, data were recorded for at least 3 min, while for the pore-spanning membrane, at least 5 min were used to build the intensity correlation curve. For planar membrane measurements with dense labeling (SLB and tense GUV), at least 20 minutes were used to smooth the correlation curve. Please note that we performed for each GUV/RBC/mitochondrion/PSM sample only one point measurement, but then repeated these measurements many times on different samples for determining mean values and standard deviations. Furthermore, we checked the mechanical stability of our setup (lateral sample drift, loss of focus) during each measurement by inspecting the recorded intensity-time traces, which were found to remain constant over the duration of each measurement (see Supplementary Fig. 9).

## Substrate preparation

The protocol for preparing the gold-modified substrate has been previously described in our publications[31,33]. Briefly, we used a layer-by-layer electron-beam evaporation process to deposit a 2 nm titanium layer, followed by a 10 nm gold layer, another 1 nm titanium layer, and finally a 10 nm SiO$_2$ layer on the surface of a glass coverslip. Slowest rate of deposition was maintained (1 Å s$^{-1}$) to ensure maximal homogeneity. The spacer thickness was continuously monitored during evaporation with an oscillating quartz unit. This gold-covered substrate is referred to as the MIET substrate. Characterizations of the AFM and scanning electron microscopy demonstrate the roughness is only 1.2 nm and the MIET substrate exhibits exceptional smoothness and uniformity (see Supplementary Fig. 10). Prior to conducting MIET experiments, the MIET substrate was subjected to plasma cleaning for 60 s at the highest plasma intensity (Harrick Plasma). Subsequently, it was incubated in 5 mg/mL$^{-1}$ bovine serum albumin (BSA, Sigma-Aldrich) solution for 15 min to prevent any non-specific interactions during experiments involving GUVs and RBCs.

For GIET, we used a transfer method based on the manufacturer's instructions (Easytransfer, Graphena. Inc.) to prepare a graphene-modified substrate. Silica with the desired thickness was then evaporated onto the surface of the graphene-modified substrate to create the GIET substrate. To prepare the pore-containing graphene substrate, we employed the nanosphere lithography method (Supplementary Fig. 4)[58]. First, we evaporated a 2 nm SiO$_2$ layer onto the graphene-modified substrate to protect the graphene film. Then, we deposited polystyrene (PS) beads with a diameter of 3 μm (Bangs Laboratories, Inc.) in water onto the SiO$_2$/graphene substrate. This substrate was then dried at room temperature. Next, silica with the desired thickness (30 nm) was evaporated on top of the PS/SiO$_2$/graphene substrate. Finally, the PS beads were removed in acetone under ultrasonication for 5 min, resulting in a substrate with pores of the desired height and a diameter of 3 μm on the graphene surface.

For the pore substrate without graphene for pore spanning membrane (PSM) measurements above the glass, we deposited the PS beads onto the coverslip and coated all layer-by-layer with 2 nm Ti, 10 nm Au, 1 nm Ti and 20 nm SiO$_2$. The gold film was used to quench the fluorescence of the supported membrane, allowing us to distinguish the PSMs from the GUV patch. Before using the substrates for membrane spanning, we modified the substrates to have a positively charged surface[55]. We first treated the substrates with plasma cleaning for 30 s at low intensity and then incubated them in a solution of 1% (v/v) (3-trimethoxysilylpropyl)-diethylenetriamine (DETA) in water for 15 min. The positively charged substrates were then washed with methanol and water, followed by heating at 110 °C for 15 min.

To prepare the graphene substrate for mitochondria measurements, we used a 2 nm or 10 nm thick SiO$_2$-spacer and treated it with 0.1% (wt/v) poly-L-lysine (PLL) solution for 5 min. The substrate was then rinsed with water and allowed to dry. The resulting positively charged substrates were stored in a clean, dust-free environment to prevent contamination.

Note that we have taken into account that the BSA layer has a thickness of 3 nm[66], and the PLL layer has a thickness of 1 nm[67] when calculating the MIET and GIET calibration curves.

## Lipids

1-stearoyl-2-oleoyl-sn-glycero-3-phosphocholine (DOPC), 1,2-dioleoyl-sn-glycero-3-phosphoethanolamine-N-(methoxy (polyethyleneglycol)-2000) (DOPE-PEG2000), (1,2-dioleoyl-sn-glycero-3-phosphoethanolamine-N-(cap biotinyl) (DOPE-cap-biotin), 1,2-dioleoyl-sn-glycero-3-phosphoethanolamine (DOPE), 1,2-dioleoyl-3-trime thylammonium-propane,chloride salt (DOTAP), 1,2-diphytanoyl-sn-glycero-3-phosphocholine (DPhPC) and 1,2-dioleoyl-sn-glycero-3-phospho-(1'-rac-glycerol) (DOPG) were purchased from Avanti Polaer Lipids and diluted to 10 mg/mL in chloroform as stock solution and stored at −20 °C. The Atto655-dyes labeled 1,2-dipalmitoyl-sn-glycero-3- phosphoethanolamine (DPPE-Atto655) were purchased from ATTO-TEC GmbH and dissolved in chloroform at a concentration of 0.01 mg/mL and 1 mg/mL for stock solution.

## Vesicles preparation

Giant unilamellar vesicles (GUVs) were produced using the electro-swollen method[33]. For MIET measurements, a lipid mixture containing SOPC with 2 mol% DOPE-PEG2000, 5 mol% DOPE-cap-biotin and 0.001-1 mol% DPPE-Atto655 was used, while for PSM measurements, a mixture containing DPhPC with 10 mol% DOPG and 0.001-1 mol% DPPE-Atto655 was used. The mixture was deposited onto an electrode and allowed to evaporate under vacuum for 3 h at 30 °C. The chamber was then filled with a sucrose solution (230 mM for MIET-GUV measurements and 300 mM for PSM measurements). An alternating electric current at 15 Hz and a peak-to-peak voltage of 1.6 followed was applied for 3 h to the chamber used for electro-formation, followed by a treatment with 8 Hz for 30 min. The final GUV suspension was stored at −4 °C and used within 3 days.

For the MIET-GUV experiments, the GUV suspension was diluted 50-folded in PBS buffer (10 mM Na$_2$HPO$_4$, 2 mM KH$_2$PO$_4$, and 3 mM

KCl, pH 7.4) with varying osmolarity by adding NaCl. Deflated GUVs were observed in PBS buffer with 400 mOsml$^{-1}$, while tense GUVs were observed in PBS buffer with 230 mOsm/L. For PSM experiments, the stock GUV suspension was diluted 50-fold in Tris-HCl buffer (20 mM Tris-HCl, 100 mM NaCl, 10 nM CaCl$_2$, pH 7.4). For all the GUV experiments, the diluted GUV solution with observation buffer was added to a Culture-Insert 4 Well chamber (ibidi GmbH) on the modified substrate. The chamber was then covered with a glass slide and kept at room temperature for 15 min to allow for equilibration before measurement.

### Preparation of RBCs

RBCs were prepared by pricking a finger and diluting the blood 100-fold with PBS buffer (osmolarity of 300 mOsm/L). The resulting suspension was collected by centrifugation at $200 \times g$ for 1 min, and the pellet was fluorescently stained using a liposome method (see next subsection). The stained pellet was then washed twice with PBS buffer and dissolved in the observation buffer. For the ATP-saturated RBC experiments, the observation buffer contained 0.1 mg/mL BSA and 10 mM D-Glucose, while for the ATP-depleted RBCs, the buffer contained only 0.1 mg/mL BSA. The RBCs were ATP-depleted by incubating them in the observation buffer overnight at 37 °C. After staining, the cells were used within one hour. The sex and/or gender were not considered in this study design because the primary focus of the study doesn't involve investigating differences related to sex or gender, conducting separate analyses might not align with the research objectives.

### Fluorescent staining of RBCs

We employed a fusogenic liposome method[21,68] to label the RBCs with a fluorescent marker. Initially, a lipid mixture comprising DOPE, DOTAP, and DPPE-Atto655 in a 1:1:0.2 weight ratio was dried under vacuum for 3 h at 30 °C and re-suspended in 20 mM Tris-HCl (pH 7.4) buffer at a concentration of 2.2 mg/mL. The suspension was vortexed for 2 min and sonicated for 10 min to generate small multilamellar liposomes. To stain the cells, RBCs were incubated in the diluted liposome solution (1:100) for 15 min at 37 °C. The resultant fluorescently labeled RBCs were then washed twice with PBS buffer (300 mOsml$^{-1}$) by centrifugation ($200 \times g$, 1 min) and re-suspended in the observation buffer.

### Isolation of mitochondria

The isolation of mitochondria from cultured MH-S cells (ATCC) was performed using the Thermo Scientific™ mitochondrial isolation kit. The isolated mitochondria were snap-frozen in liquid nitrogen and stored at −80 °C in 300 mM trehalose buffer (300 mM trehalose, 10 mM HEPES-OH, 10 mM KCl, 1 mM EGTA, and 0.1% BSA, pH 7.7). This buffer has been reported to preserve the biological functions of mitochondria and maintain the integrity of their outer membrane[69].

### Mitochondria membranes measurement

In a typical measurement of mitochondria, 5 µL of frozen mitochondria were thawed by holding the tube between fingers and mixed with 25 µL of trehalose buffer. The resulting solution was added on top of the PLL-modified graphene substrate with a 2 nm SiO$_2$ spacer for IMM or a 10 nm SiO$_2$ spacer for OMM measurements. The plates were incubated at 4 °C for 30 min, followed by the addition of 70 µL of respiration buffer[63] (130 mM KCl, 5 mM K$_2$HPO$_4$, 20 mM MOPS, 1 mM EGTA, 0.1% (w/v) BSA, pH 7.1) containing MitoTracker™ Deep Red (500 nM) or CellMask™ Deep Red (10 µg/mL). After 15 minutes of incubation at room temperature, the buffer was carefully changed to a new respiration buffer without dye. For ADP stimulated respiration measurements, we added 1 mM malic acid, 5 mM pyruvate, and 100 µM ADP to the respiration buffer. The mitochondria were used for up to one hour after fluorescent staining.

### Statistics & reproducibility

All values are expressed as the mean ± SD. No statistical method was used to predetermine sample size, and the sample size was chosen based on similar studies published before[15,21]. No data were excluded from the analyses. All experiments were repeated at least twice with reproducible results. In this study, randomization wasn't applicable in the traditional sense as there was no random assignment of participants to different groups or treatments. Instead, the focus was on analyzing measurements or samples taken from a single healthy source (red blood cell measurements). For the measured red blood cells samples, these measurements were randomized by randomly collecting different individual cells from the blood sample. To minimize bias, the study did not involve assigning individuals to different experimental conditions. The Investigators were not blinded to allocation during experiments and outcome assessment because it involved analyzing red blood cells from a single healthy individual, without any treatments or interventions administered. In this case, there are no varying conditions or interventions to blind participants or researchers.

### Reporting summary

Further information on research design is available in the Nature Portfolio Reporting Summary linked to this article.

## Data availability

Source data are provided with this paper. There are no restrictions on data availability. All data supporting the findings of this study are available within the main text, supplementary information, and 'Source data' Excel file. Two unprocessed raw data files are also deposited in the GitHub database at https://gitlab.gwdg.de/igregor/miet-membrane-fluctuation.git. Upon request, all raw data files can be requested from the first author Dr. Tao Chen (tao.chen@phys.uni-goettingen.de). Requests will be fulfilled within 4 weeks. Source data are provided with this paper.

## Code availability

All code used for generating Figs. 1–5 are deposited on GitHub at https://gitlab.gwdg.de/igregor/miet-membrane-fluctuation.git. In particular, the depository contains: • Matlab (v. 2022b, MathWorks® Inc.) code used for calculating the model curves of Figs. 1c, d, 2d–g, 3c, 4c and 5c and for fitting the deflated GUV and ATP-depleted RBC height correlation curve in Figs. 2g and 3c; • A Mathematica (v. 13.2.1.0 Wolfram Research Inc.) notebook that generates the graphs of all figures; • A ReadMe file in the depository to explain all the analysis details. The FLIM images (Figs. 2b, c, 3b, 4b, 5a, b) are analyzed by using a published software (TrackNTrace Lifetime Edition)[70], Additionally, all codes are citable from Zenodo (DOI: 10.5281/zenodo.10473902)[71].

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

## Acknowledgements
T.C. and J.E. acknowledge financial support by the European Research Council (ERC) for financial support via project "smMIET" (grant agreement no. 884488) under the European Union's Horizon 2020 research and innovation program. J.E. acknowledges financial support by the DFG through Germany's Excellence Strategy EXC 2067/1-390729940. We thank Anna Chizhik, Ingo Gregor, and Alexey Chizhik for their help and support. We acknowledge support by the Open Access Publication Funds/transformative agreements of the Göttingen University.

## Author contributions
T.C. prepared all samples, performed all the measurements. T.C. and N.K. and analyzed the data. J.E. made all figures in the main text. All authors helped with preparing the manuscript, which was written by J.E.

## Funding

## Competing interests
The authors declare no competing interests.
