## [Peer Review File · Nature Communications]

REVIEWER COMMENTS

Reviewer #1 (Remarks to the Author)

The investigation of membrane fluctuations is crucial in understanding the physical properties and functional mechanisms of biological membranes. Membrane fluctuations play a key role in critical processes, including membrane fusion, protein-lipid interactions, and signal transduction. The evaluation of out-of-plane membrane fluctuations remains a very challenging task. This study proposes a new and highly effective tool that combines FCS with metal-induced energy transfer (MIET) to address the urgent need for precise measurements of these fluctuations.

The methodological approach presented in this publication is remarkable and represents a significant advancement. However, there are a few minor comments that need to be addressed. Firstly, it would be beneficial to provide further clarification on the structure of the giant unilamellar vesicles (GUVs) on the substrate. Specifically, it is important to comprehend whether the GUVs spread, take on a spherical shape, or undergo deflation, as such characteristics could impact the subsequent analysis. Moreover, it is suggested that the projected area of the GUVs be included as supplementary information, as this would augment the comprehensiveness of the findings.

Further, the precision and reproducibility of MIET measurements can certainly be greatly impacted by the quality and properties of the gold surface. A comprehensive analysis of the gold surface, including evaluations of its roughness, uniformity, and the existence of flaws or impurities, is crucial to guarantee the dependability of experimental results. Therefore, it is desirable to perform a thorough characterization of the gold surface, using techniques like atomic force microscopy (AFM) or scanning electron microscopy (SEM), to confirm its appropriateness for enabling precise MIET measurements, as supplementary information.

The exceptional paper with an innovative methodology to address a relevant inquiry in the field of membrane biophysics and beyond, has far-reaching implications in diverse scientific disciplines and is undoubtedly worthy of publication.

Reviewer #2 (Remarks to the Author)

It was a pleasure to read about the next application of MIET, developed in the lab of prof. Enderlein, where Chen et al. couple it with FCS to quantify the amplitude and frequency of membrane undulations. To observe emission intensity oscillations from varying distances of the probes to the metal or graphene layer, they neatly eliminate the influence of the standard fluctuations from probes' in-plane diffusion by

employing high concentrations of the dye. They demonstrate the use of the technique on several artificial and cell-derived membranes.

The experiments were performed and analysed with care, the data are nicely presented, and the text is in most parts clearly written. I will be able to recommend the publication of this work after a minor revision, please address the following points:

First please elaborate on the following points about the methodology:

1. (p.5) The theory assumes “averaging over all possible [dipole] orientations”. This is likely not the case for membrane-bound probes that often show considerable anisotropy. Please discuss the consequences of this assumption.
2. (p.6/Supp Fig 1) Were parameters (τ , QE, ..) determined by fitting MIET-FCS data or taken from other experiments?
3. (p.7) Was there any specific reason for using SOPC over more widely used POPC in GUVs?
4. (p.7 and beyond) Please specify whether the +/- values refer to the standard deviation or anything else.
5. (p.8) Where is the h_0 measured from? The substrate, metal/graphene layer, last glass surface, BSA coating, ... ?
6. (p.8) As deflation, lipid composition etc. greatly affect the fluctuations, please specify the conditions for studies you get “excellent agreement” with.
7. (p.11, Fig 3) The ATP- curve seems to contain two decay components. Is this consistent across all measurements? Which part was included in the fit?
8. (p.12/Supp Fig 5a) How do you know there is a pore under the bright spot?
9. (p.12) When your data are not converted into bending rigidity and other material constants, how can one compare your τ^* values with other methods’ outputs (e.g. power spectrum from fluctuation spectroscopy)?
10. (p.13) Please explain the rationale behind different substrate thickness used for OMM and IMM – was it to obtain similar effect for both conditions?
11. (p.14) Do you consider the small decrease in height for IMM between ADP+/- significant?
12. (p.14) “consistent with previous reports using static GIET measurements” – Doesn’t the height come from a static lifetime measurement here too?

Then I recommend some improvements to the graphical material:

13. (p.7/Fig 2d) For 1000-fold higher concentration, a 1000-fold lower ACF amplitude is expected; in the current Fig2d, the same scaling of the y-axis for both concentrations prevents one to evaluate the absence of correlation for the higher concentration; please guide the reader to panel e if it shows the same dataset, or provide a rescaled display for the orange curve in panel d.

14. (p. 11, Fig 3) To make the final information better visible, the parameters' histograms could benefit from larger fonts and more space in the figures.

15. (p.12/Supp Fig 5f) Please check if the red curve, acquired over a pore, is appropriately labelled as “t-GUV”.

16. (p.13, Fig 4a) Please indicate the “two types of pores” on the image.

Please also consider the following suggestions to improve the text:

17. (p.1) Mentioning membrane “undulations” in the title or abstract may help attract additional readership.

18. (p.2) Several fact-claiming sentences end without a reference (e.g. “Brochard and Lennon presented ...”, “... one can evaluate ...”).

19. (p.2) Claims about “far-reaching implications” for “improved human health” are a bit of a stretch. If no references for at least half-way applications are given, the expectations should be toned down a bit – the work is nice already in its scope of optics and biophysics. (Comment applies to discussion/conclusion as well.)

20. (p.3) At points it gets a bit muddled how this approach fills a gap in the previous methodology. “... current state-of-the-art techniques do not achieve the necessary resolution ... ” seemingly contradicts the previous sentence referencing a work with high-resolution results. Statements like “[MIET/GIET-FCS] is a significant improvement over previous techniques” should be substantiated or left out.

21. (p.8) Many readers may not be familiar with DODS - please provide a brief explanation why sub-focusing is used there.

There are also some small hitches in text, some of which would likely be fixed in the typesetting phase anyway:

22. (p.1) “microsecond that” >> “microsecond, which”

23. (p.5) “glass over slide” – do you mean coverslip or something else?

24. (p.5) “680,nm”

25. (p.7, last sentence of chapter 2) Is there anything missing in the part about tau*?

26. (p.8) “an correlation”

27. (p.8) Units of osmolarity are likely mOsm per L, not mL.

28. (p.12, 3rd paragraph) Check if Supp Fig 6 was really meant here.
29. (p.12) Were the pores really “etched” out of the substrate? The figure and methods suggest a different procedure.
30. (p.12) “small ... lifetime” >> “short ... lifetime”
31. (p.12 and elsewhere) Some missing spaces between numbers and units

Finally, please discuss some exciting follow-up questions that this work brings up:

32. (p.11) What could be the reason for faster dissipation of fluctuations in ATP+ cells?
33. (p.12) Membrane heights over the pores are at about half the nominal depth - any idea why?
34. (p.12) It would be instructive to learn why/how the standard mechanics model fails to fit the data.
35. (p.14) How does the convoluted geometry of the IMM affect the fluctuations and their measurements?
36. Could a similar approach work for measuring nano-topography of membranes pinned to the substrate (say by receptors)?

Looking forward to reading the revised version.

Iztok Urbancic

Reviewer #3 (Remarks to the Author)

The paper describes membrane fluctuation measurements with high axial and temporal resolution based on metal-/graphene-induced energy transfer (MIET/GIET), first introduced in 2005/2016. MIET/GIET relies on the interaction between a fluorescent emitter and surface plasmons in metal or excitons in graphene. This interaction affects the fluorescent dye's excited state lifetime as well as brightness. By measuring this lifetime/brightness, one can determine the distance between the dye and the metal or graphene layers.

Specific comments:

1. The MIET/GIET method requires labeling, which would involve dedicated protocols for sample preparation. Does fluorophore density affect MIET/GIET measurement sensitivity? If so, how can one optimize sample preparation?

2. On a related note, the biological sample would not be in its native state once tagged with an exogenous marker. How does it impact the ability to study live cells/specimens?

3. Photobleaching is another concern. In the manuscript, the authors describe, "For GUV, mitochondrion, and cell measurements, data were recorded for at least 3 minutes, while for the pore-spanning membrane, at least 5 minutes were used to build the intensity correlation curve. For planar membrane measurements with dense labeling (SLB and tense GUV), at least 20 minutes were used to smooth the correlation curve." However, the authors did not discuss if photobleaching played any role during minutes long measurements at single locations.

4. Figure 3(b) was not explained. Given that it's a single point measurement technique with confocal detection diameter of 280 nm, did the researchers make multiple measurements along the rim formed between the RBC and gold-plated substrate? If so, how many measurements were made. How long did each measurement take / what was the total time? How did the researchers assure that the sample did not drift laterally or axially during the experiment?

5. AFM is another point measurement technique that allows membrane fluctuation measurements with sub-nanometer scale sensitivity. What are the advantages / disadvantages of the current technique (MIET/GIET) compared to AFM? Is one always better than other or are there specific situations where one could benefit from one versus other? Please highlight.

6. For the ATP-related measurements in RBCs, it is recommended that the researchers use measured height correlations [Fig. 3(c)] / brightness to determine biomechanical parameters such as membrane tension and compare the findings with prior work based on DPM and AFM.

7. MIET/GIET is essentially a single-point measurement technique. While the approach promises high temporal and axial motion sensitivity, correlation with other lateral locations wouldn't be possible. How does this limitation compare with optical techniques such as diffraction phase microscopy (DPM), which can easily make wide-field measurements at > 100 Hz?

8. On page 11 of the manuscript, it is mentioned that the dynamic range of MIET/GIET is 160/20 nm. This makes this technique suitable only for basal surfaces. Literature <https://opg.optica.org/ol/abstract.cfm?URI=ol-39-20-6062> indicates that top and bottom surfaces of RBC

display different thermal fluctuations due to close contact with the surface on which the cell is resting. The very nature of this approach will measure only subdued membrane fluctuations in intact biological cells. Please comment.

9. How can one use this approach to quantify fluctuations of top or sub-cellular surfaces such as nuclear membrane in intact cells.

10. On page 15, the statement “This technique holds great promise for assessing cellular functioning in response to drugs and therapies, providing insights into health, mechanical properties, and treatment responses for improved healthcare outcomes. Our method has vast potential for applications in the study of biological membranes both in vitro and in vivo.” is a very general/broad statement. The authors should be more specific and make more realistic claims. For example, in vivo studies with a dynamic range of 160/20 nm (MIET/GIET) is a bit of a stretch.

Response to the reviewer's comment to our manuscript "Measuring sub-nanometer fluctuations at microsecond temporal resolution with metal- and graphene-induced energy transfer spectroscopy" by Tao Chen, Narain Karedla, and Jörg Enderlein

REVIEWER COMMENTS

Reviewer #1 (Remarks to the Author)

The investigation of membrane fluctuations is crucial in understanding the physical properties and functional mechanisms of biological membranes. Membrane fluctuations play a key role in critical processes, including membrane fusion, protein-lipid interactions, and signal transduction. The evaluation of out-of-plane membrane fluctuations remains a very challenging task. This study proposes a new and highly effective tool that combines FCS with metal-induced energy transfer (MIET) to address the urgent need for precise measurements of these fluctuations. The methodological approach presented in this publication is remarkable and represents a significant advancement.

We thank the reviewer for her/his comments that helped us to improve our manuscript.

However, there are a few minor comments that need to be addressed. Firstly, it would be beneficial to provide further clarification on the structure of the giant unilamellar vesicles (GUVs) on the substrate. Specifically, it is important to comprehend whether the GUVs spread, take on a spherical shape, or undergo deflation, as such characteristics could impact the subsequent analysis. Moreover, it is suggested that the projected area of the GUVs be included as supplementary information, as this would augment the comprehensiveness of the findings.

To clarify the structure of the GUV on the substrate, we scanned the GUV along the xz plane and found that the GUV adopts a non-spherical shape (see new Supplementary Figure 11). We did also determine the projected area for 25 GUVs. We added one xz image and a histogram of the projected area distribution to the revised supplementary information (see new Supplementary Figure 11).

Further, the precision and reproducibility of MIET measurements can certainly be greatly impacted by the quality and properties of the gold surface. A comprehensive analysis of the gold surface, including evaluations of its roughness, uniformity, and the existence of flaws or impurities, is crucial to guarantee the dependability of experimental results. Therefore, it is desirable to perform a thorough characterization of the gold surface, using techniques like atomic force microscopy (AFM) or scanning electron microscopy (SEM), to confirm its appropriateness for enabling precise MIET measurements, as supplementary information.

MIET substrates were prepared by evaporation using an electron beam source. We maintained the slowest rate of deposition (1 \AA s^{-1}) to ensure maximal homogeneity of the deposited layer. During evaporation, the spacer thickness was continuously monitored with an oscillating quartz unit. We have performed additional AFM and SEM measurements to estimate the quality of the obtained MIET substrates, and we found that the roughness of the silica surface is only 1.2 nm (mean square root) showing that the surface is very uniform. We added a corresponding paragraph to the Method section in the new main text, and added Supplementary Figure 10 showing AFM and SEM images of the MIET substrate to the Supplementary Information.

The exceptional paper with an innovative methodology to address a relevant inquiry in the field of membrane biophysics and beyond, has far-reaching implications in diverse scientific disciplines and is undoubtedly worthy of publication.

Reviewer #2 (Remarks to the Author)

It was a pleasure to read about the next application of MIET, developed in the lab of prof. Enderlein, where Chen et al. couple it with FCS to quantify the amplitude and frequency of membrane undulations. To observe emission intensity oscillations from varying distances of the probes to the metal or graphene layer, they neatly eliminate the influence of the standard fluctuations from probes' in-plane diffusion by employing high concentrations of the dye. They demonstrate the use of the technique on several artificial and cell-derived membranes.

The experiments were performed and analysed with care, the data are nicely presented, and the text is in most parts clearly written. I will be able to recommend the publication of this work after a minor revision, please address the following points:

We thank the reviewer for his comments that helped us to improve our manuscript.

First please elaborate on the following points about the methodology:

1. (p.5) The theory assumes “averaging over all possible [dipole] orientations”. This is likely not the case for membrane-bound probes that often show considerable anisotropy. Please discuss the consequences of this assumption.

We performed additional fluorescence anisotropy measurements on GUV membranes and found the dye exhibits random orientation in deflated GUV (see new Supplementary Figure 8). We have added the figure of the anisotropy images in the new Supplementary Information.

2. (p.6/Supp Fig 1) Were parameters (τ , QE, ..) determined by fitting MIET-FCS data or taken from other experiments?

The quantum yield and free-space fluorescence lifetime were determined in our previous paper where we determined the quantum yield of membrane associated dyes: Schneider et al., J. Phys. Chem. Lett. 2017, 8, 7, 1472–1475. We have added this citation to the Supplementary Information.

3. (p.7) Was there any specific reason for using SOPC over more widely used POPC in GUVs?

We used SOPC because this lipid is intensely used in many previous studies relating to membrane fluctuation (e.g. Fenz et al., Nature Phys. 2017, 13, 906–913; Monzel et al., Nature Commun. 2015, 6, 8162; Betz et al., Soft Matter, 2012, 8, 5317-5326; Monzel et al., Soft Matter, 2016, 12, 4755-4768; Schmidt et al., Phys. Rev. X 2014, 4, 021023).

4. (p.7 and beyond) Please specify whether the +/- values refer to the standard deviation or anything else.

It refers to standard deviation. We have specified this in the revised manuscript.

5. (p.8) Where is the h_0 measured from? The substrate, metal/graphene layer, last glass surface, BSA coating, ... ?

For GUV and red blood cell measurements, h_0 is the height measured from the BSA surface as determined with MIET. For the mitochondria measurements, h_0 is the height from the PLL surface. Here, we assumed that the BSA monolayer has a 4 nm thickness (Sarkar and Kundu, JCI Open, 2021, 3, 100016), and the PLL monolayer a 1 nm thickness (Kosior et al., J. Phys. Chem. C, 2020, 124, 8, 4571–4581). These values were taken into account when calculating the MEIT and GIET curves. We added this information to the Method section in the main text.

6. (p.8) As deflation, lipid composition etc. greatly affect the fluctuations, please specify the conditions for studies you get “excellent agreement” with.

Indeed, we compared our results with published studies on DMPC (FLIC) and SOPC (RICM). Interestingly, even when comparing our SOPC results with published results on DMPC, we find a very similar behavior. We have rephrased this sentence to: “This is in excellent agreement with published values obtained for SOPC using RICM ($h_0 = 31\text{--}39$ nm and $\psi = 4\text{--}15$ nm) [47]. Interestingly, our SOPC results are very similar to published results on DMPC using FLIC microscopy ($h_0 = 30\text{--}60$ nm and $\psi = 10$ nm) [15, 23, 24].”

7. (p.11, Fig 3) The ATP- curve seems to contain two decay components. Is this consistent across all measurements? Which part was included in the fit?

Indeed, the shown curve seems to show a double decay. We have performed additional RBC measurements and found in almost all cases a mono-exponential decay behavior. Thus, our original figure was not representative, and we have exchanged it with a more typical measurement result, see Fig 3. In the original version of our manuscript, we did not fit the RBC data, which is now done.

8. (p.12/Supp Fig 5a) How do you know there is a pore under the bright spot?

It is important to emphasize that in Supplementary Figure 5a, the bottom of the pore is the bare glass substrate, while a 10 nm thick gold layer is present in the non-porous regions. Consequently, the fluorescence from the membrane on the non-porous regions is quenched by the gold layer. However, from the membrane parts suspended over the pores, fluorescence remains unquenched. Moreover, the known pore size (3 μm) and size of the bright circular patches in the fluorescence image are identical, which also supports the interpretation of the bright spots to be pore-spanning membranes. Additionally, we measured the diffusion of dye-labeled lipids in the bright spots with FCS and found diffusion values consistent with the lipid diffusion in free-standing membranes, much faster than the diffusion in supported membranes. We added a corresponding clarification to the Supplementary Information.

9. (p.12) When your data are not converted into bending rigidity and other material constants, how can one compare your τ^* values with other methods’ outputs (e.g. power spectrum from fluctuation spectroscopy)?

Due to the complexity of the mechanical behavior of the PSM above a membrane-closed pore, we cannot determine the bending rigidity or other material constants for the PSMs. This would require extensive numerical hydrodynamics/membrane-mechanics modeling which goes far beyond the scope of our manuscript. However, we are convinced that our measured values of the fluctuation relaxation times will be valuable for future theoretical studies of membrane behavior over closed pores.

10. (p.13) Please explain the rationale behind different substrate thickness used for OMM and IMM – was it to obtain similar effect for both conditions?

The thickness of the silica spacer we used for the OMM is 10 nm, while for the IMM it is 2 nm. This choice was made because the working distance of GIET is in the range of approximately 5–30 nm above the graphene surface. The ideal working region is around 10 to 20 nm where the slope of the lifetime-versus-distance curve is steepest. We wanted to ensure that both the OMM and IMM are located within

this optimal working region. Thus, for the IMM, which is farther away from the surface (approximately 15 nm from the PLL surface), we used a thinner spacer. This added this clarification to the revised manuscript.

11. (p.14) Do you consider the small decrease in height for IMM between ADP+/- significant?

Yes, the height decrease of IMM in the presence of ADP is directly related to the biological function of the IMM: it signifies that the distance between IMM and OMM becomes smaller (OMM heights do not change in these two states), which should improve molecule transport and exchange between IMM and OMM. Similar observations have been reported by using electron microscopic tomography (Frey et al., Biochimica et Biophysica Acta, 2002, 1555, 196–203). We added to the main text the sentence: “The reduced distance between the OMM and IMM in the active state enhances transport and exchange of molecules between these two membranes. A similar observation has been reported using electron microscopic tomography [63].”

12. (p.14) “consistent with previous reports using static GIET measurements” – Doesn’t the height come from a static lifetime measurement here too?

We completely agree: Our reported height values are time-averaged values which average out any membrane fluctuations. The resulting mean fluorescence lifetime then yields the mean height value. In this sense, one can call it also a static lifetime measurement. To prevent any confusion, we deleted “using static GIET measurements”.

Then I recommend some improvements to the graphical material:

13. (p.7/Fig 2d) For 1000-fold higher concentration, a 1000-fold lower ACF amplitude is expected; in the current Fig2d, the same scaling of the y-axis for both concentrations prevents one to evaluate the absence of correlation for the higher concentration; please guide the reader to panel e if it shows the same dataset, or provide a rescaled display for the orange curve in panel d.

We have included a zoom-in into Figure 2d to illustrate the weak correlation in the case of high labelling density. We amended the figure caption correspondingly.

14. (p. 11, Fig 3) To make the final information better visible, the parameters' histograms could benefit from larger fonts and more space in the figures.

We have changed Fig. 3 as requested.

15. (p.12/Supp Fig 5f) Please check if the red curve, acquired over a pore, is appropriately labelled as “t-GUV”.

The red curve in the original Supplementary Figure 5f is the FCS curve of t-GUV. We have changed the curve color in Figure 5f, so it's consistent with the Supplementary Figure 5a and 5b now.

16. (p.13, Fig 4a) Please indicate the “two types of pores” on the image.

We added annotation to Fig. 4b and amended the figure caption.

Please also consider the following suggestions to improve the text:

17. (p.1) Mentioning membrane “undulations” in the title or abstract may help attract additional readership.

We now mention “undulations” in the title.

18. (p.2) Several fact-claiming sentences end without a reference (e.g. “Brochard and Lennon presented ...”, “... one can evaluate ...”).

We have added citations to all such statements.

19. (p.2) Claims about “far-reaching implications” for “improved human health” are a bit of a stretch. If no references for at least half-way applications are given, the expectations should be toned down a bit – the work is nice already in its scope of optics and biophysics. (Comment applies to discussion/conclusion as well.)

We have deleted this statement.

20. (p.3) At points it gets a bit muddled how this approach fills a gap in the previous methodology. “... current state-of-the-art techniques do not achieve the necessary resolution ...” seemingly contradicts the previous sentence referencing a work with high-resolution results. Statements like “[MIET/GIET-FCS] is a significant improvement over previous techniques” should be substantiated or left out.

We have rephrased this sentence to lift the confusion: “Nevertheless, as of now, there have been no reports uncovering these specific fluctuations.”

21. (p.8) Many readers may not be familiar with DODS - please provide a brief explanation why sub-focusing is used there.

We have added the following short explanation of DODS to the main text: “In DODS, the membrane is positioned in such a way that its mean position is located at the inflection point of the excitation intensity profile. This leads to measurable intensity fluctuations when the membrane undergoes strong bending fluctuations.”

There are also some small hitches in text, some of which would likely be fixed in the typesetting phase anyway:

We thank the reviewer for pointing out these hitches. We have corrected them all, and made also the following changes.

22. (p.1) “microsecond that” >> “microsecond, which”

We have corrected that.

23. (p.5) “glass over slide” – do you mean coverslip or something else?

It is a glass coverslip, we have corrected that.

24. (p.5) “680,nm”

We have corrected that.

25. (p.7, last sentence of chapter 2) Is there anything missing in the part about τ^* ?

After „ τ^ “ we have now added „which is determined by the point where this function has decreased to one-half of its maximum value.“ for clarification.*

26. (p.8) “an correlation”

We have corrected that.

27. (p.8) Units of osmolarity are likely mOsm per L, not mL.

We have corrected all both in the new main text and the Supplementary Information.

28. (p.12, 3rd paragraph) Check if Supp Fig 6 was really meant here.

We have deleted “see Supplementary Figure 6” from the main text.

29. (p.12) Were the pores really “etched” out of the substrate? The figure and methods suggest a different procedure.

We have rephrased this sentence and cited an additional paper to make it clearer.

30. (p.12) “small ... lifetime” >> “short ... lifetime”

We have corrected that.

31. (p.12 and elsewhere) Some missing spaces between numbers and units

We have corrected that.

Finally, please discuss some exciting follow-up questions that this work brings up:

32. (p.11) What could be the reason for faster dissipation of fluctuations in ATP+ cells?

It should be pointed out that from our result, the ATP+ RBCs show a slower dissipation of fluctuations (larger tau) than the ATP- RBCs. This observation is consistent with published DODS measurements (Monzel et al., Nature Commun. 2015, 6, 8162), and it is also seen in optical tweezer measurements (Turlier et al., Nature Phys. 2016, 12, 513–519). It is assumed that active ATP-powered processes result in a slower-than-thermal fluctuation dynamics. The observed slower dissipation dynamics of membrane fluctuations could result from the active motion of the spectrin network (Turlier et al., Nature Phys. 2016, 12, 513–519). We extended the discussion of this point in the main text.

33. (p.12) Membrane heights over the pores are at about half the nominal depth - any idea why?

Similar observations for PSMs prepared from GUVs have been reported before using iSCAT (Susann et al. Nano Lett. 2018, 18, 8, 5262–5271), ion-conductance microscopy (SICM, see e.g. Schütte et al., PNAS, 2017, 114, E6064-E6071; Höfer et al., Soft Matter, 2011, 7, 1644-1647), and AFM (Simon et al., Biochimica et Biophysica Acta, 2013, 1828, 2739–2744). A comprehensive theoretical study of the mechanics of PSMs has been published in: Mey et al., J. Am. Chem. Soc. 2009, 131, 20, 7031–7039. There it was shown that membrane bending is strongly dependent on surface functionalization and lipid composition. For a hydrophilic surface, the hydrophilic rim induces a strain on the suspended bilayer. The strain depends on the hydrophilicity of the substrate. If the surface is hydrophilic (equal to water), there is no pre-stress between the membrane on the substrate and the free membrane above the pore (no bending). However, our substrate with reduced hydrophilicity (lower than that of water) induces a bending of the PSM, lowering the distance of the free-standing membrane from the surface. We have added a short explanation and the above citations to the main text of the revised manuscript.

34. (p.12) It would be instructive to learn why/how the standard mechanics model fails to fit the data.

The standard fitting model is applicable to unbound membranes, where bending waves can propagate towards infinity. However, in the case of PSMs, free wave propagation is hindered by the edge of the pore. Furthermore, the enclosed liquid in the pore does additionally dampen membrane fluctuations. This all makes a theoretical description of PSM membrane mechanics much more complicated, and no model for this situation was so far described in the literature.

35. (p.14) How does the convoluted geometry of the IMM affect the fluctuations and their measurements?

When performing a point measurement with our confocal microscope, the observed signal is always a spatial average over the size of the excitation focus. Thus, in case of the IMM of a mitochondrion, our results represent averages over lateral regions of ca. 300 nm, thus averaging out the highly convoluted geometry of the IMM. We added a corresponding explanation to the main text.

36. Could a similar approach work for measuring nano-topography of membranes pinned to the substrate (say by receptors)?

Yes, with the high axial resolution of MIET/GIET, it could be feasible to measure even Angstrom-size variations in nano-topography of substrate-supported membranes. We have added on sentence mentioning this in the discussion.

Looking forward to reading the revised version.

Iztok Urbancic

Reviewer #3 (Remarks to the Author)

The paper describes membrane fluctuation measurements with high axial and temporal resolution based on metal-/graphene-induced energy transfer (MIET/GIET), first introduced in 2005/2016. MIET/GIET relies on the interaction between a fluorescent emitter and surface plasmons in metal or excitons in graphene. This interaction affects the fluorescent dye's excited state lifetime as well as brightness. By measuring this lifetime/brightness, one can determine the distance between the dye and the metal or graphene layers.

We thank the reviewer for her/his comments that helped us to improve our manuscript.

Specific comments:

1. The MIET/GIET method requires labeling, which would involve dedicated protocols for sample preparation. Does fluorophore density affect MIET/GIET measurement sensitivity? If so, how can one optimize sample preparation?

It's important to clarify that MIET/GIET measurements are based on fluorescence lifetime rather than intensity, and therefore, fluorophore density does not significantly affect MIET/GIET results (besides accuracy which is always signal-strength dependent). However, to suppress lateral-diffusion related contributions to the correlation function, we have used in our studies rather large label concentrations. However, this is not absolutely necessary for applying MIET-GIET for measuring membrane fluctuations, it only simplifies the final data analysis and improves signal to background ratio. Otherwise, membrane labeling is usually much simpler (achieved by adding dyes or dye-labeled lipids to the solution) than protein or other biomolecular labeling, which typically requires conjugation chemistry or genetic protein modification.

In more detail, for artificial membranes (GUV and PSM), fluorophore concentration can be adjusted by mixing desired amounts of labeled lipids with normal lipids. When working with RBCs, we followed published methods, such as those described in Monzel et al., Nature Commun. 2015, 6, 8162, and Csiszar et al., Bioconjugate Chem., 2010, 21, 537–543. For the inner (IMM) and outer mitochondrial membranes (OMM), we used a commercial dye and followed the manufacturer's labeling protocol.

To make this all more clear, we added to the discussion of the main text the paragraph: "It should be emphasized that MIET/GIET measurements are based on fluorescence lifetime rather than intensity, and therefore, are much more robust with respect to labeling density and fluorescence intensity than purely intensity-based fluorescence methods. In our studies presented here, we suppressed lateral-diffusion related contributions to the correlation function by using rather large label concentrations. However, this is not absolutely necessary for applying MIET-GIET for measuring membrane fluctuations, it only simplifies final data analysis and improves signal to background ratio."

2. On a related note, the biological sample would not be in its native state once tagged with an exogenous marker. How does it impact the ability to study live cells/specimens?

This is a general problem of ALL fluorescence-based methods in microscopy and biophysics, and we certainly do not have a final solution to this problem. However, MIET/GIET does work down to the single-molecule level, which makes it as sensitive as it can possibly be. We agree that any exogenous marker may influence the activity of live cells, but a systematic study of this issue goes far beyond the scope of our current manuscript. Given that the labeling methods used in our manuscript have been employed in numerous other studies and are widely accepted, we assume that they do not significantly bias the result of our investigation.

3. Photobleaching is another concern. In the manuscript, the authors describe, "For GUV, mitochondrion, and cell measurements, data were recorded for at least 3 minutes, while for the pore-spanning membrane, at least 5 minutes were used to build the intensity correlation curve. For planar membrane measurements with dense labeling (SLB and tense GUV), at least 20 minutes were used to smooth the correlation curve." However, the authors did not discuss if photobleaching played any role during minutes long measurements at single locations.

In our measurements, the utilization of high labeling concentrations allows us to use much lower excitation power as compared to conventional methods (at least 100 times lower), effectively eliminating photobleaching in all our measurements. To provide additional evidence, we have included a supplementary figure, demonstrating no intensity decay over time during our measurements (see new Supplementary Figure 9).

4. Figure 3(b) was not explained. Given that it's a single point measurement technique with confocal detection diameter of 280 nm, did the researchers make multiple measurements along the rim formed between the RBC and gold-plated substrate? If so, how many measurements were made. How long did each measurement take / what was the total time? How did the researchers assure that the sample did not drift laterally or axially during the experiment?

In the figure caption of Figure 3, we wrote "(b) Fluorescence lifetime image of a RBC on a MIET substrate. Lifetime is shown by color, fluorescence intensity by brightness. The cell's rim touching the surface is visible as a ring of enhanced fluorescence intensity and reduced lifetime."

For one cell, we did not perform multiple measurements along the rim but only for one position along the rim, but we repeated these measurements for many different cells. Each of these measurements lasted for more than 3 minutes. We have added a note in the Method Section: "Please note that we performed for each GUV/RBC/mitochondrion/PSM sample only one point measurement, but then repeated these measurements many times on different samples for determining mean values and standard deviations." Furthermore, any lateral sample drift or loss of focus during a measurement would be readily detectable from the measured intensity-time traces, which were checked to be constant over the duration of each measurement. In the Methods section, we added the corresponding sentence "Furthermore, we checked the mechanical stability of our setup (lateral sample drift, loss of focus) during each measurement by inspecting the recorded intensity-time traces, which were found to remain constant over the duration of each measurement."

5. AFM is another point measurement technique that allows membrane fluctuation measurements with sub-nanometer scale sensitivity. What are the advantages / disadvantages of the current technique (MIET/GIET) compared to AFM? Is one always better than other or are there specific situations where one could benefit from one versus other? Please highlight.

AFM is indeed a powerful technique with high spatial resolution for studying biological surfaces. However, due to mechanical reasons, AFM is usually not suitable for tracking fast membrane fluctuations (Monzel and Sengupta, J. Phys. D: Appl. Phys., 2016, 49, 243002). One of the notable advantages of AFM is its ability to measure mechanical properties of biological samples, such as stiffness, viscoelasticity, hardness, and adhesion. Additionally, AFM is the only technique capable of measuring Young's modulus of biological membranes (Tomaiuolo, Biomicrofluidics, 2014, 85, 051501). However, AFM cannot measure membrane tension. Complementary, MIET/GIET-FCS offers the advantage of non-invasive detection and high time resolution, allowing to monitor and quantify membrane fluctuations, making it suitable for retrieving membrane properties like tension, fluctuation amplitudes, or fluctuation times. Thus, both AFM and MIET/GIET-FCS have their specific strengths and limitations, and can be considered as complementary techniques. We have added the following paragraph to the main text:

"Atomic force microscopy (AFM) is another powerful technique for studying the mechanical properties of biological materials, including stiffness, viscoelasticity, hardness, and adhesion. However, AFM is usually not suitable for tracking fast membrane fluctuations [15]."

6. For the ATP-related measurements in RBCs, it is recommended that the researchers use measured height correlations [Fig. 3(c)] / brightness to determine biomechanical parameters such as membrane tension and compare the findings with prior work based on DPM and AFM.

We have fitted the ATP-depleted data following the reviewer's suggestion. In the revised manuscript, we added more RBC measurements and fitted the ATP-depleted (ATP-) data for determining tension (σ) and bending modulus (k) using equilibrium theory. We did not fit the ATP+ data because equilibrium theory is not suitable for describing actively driven membrane fluctuations (Turlier et al., Nature Phys., 2016, 12, 513–519).

We obtained values of tension $2.7 \pm 2.1 \mu\text{J}/\text{m}^2$ ($N = 22$) and bending modulus $(3.9 \pm 2.7) \times 10^{-20} \text{ J}$, and we have added a table to the Supplementary Information which compares these values with those obtained by other techniques (see Supplementary Table 1). Although for different methods, these values show huge variations, our results are on the same order as published values.

To the main text, we added the paragraph: "We used eq. 4 to fit the height correlation functions for ATP-depleted RBCs, see also Supplementary Note 4, yielding values for membrane tension ($2.7 \pm 2.1 \mu\text{J}/\text{m}^2$) and bending modulus ($(3.9 \pm 2.7) \times 10^{-20} \text{ J}$). Our results are in fair agreement with published values using other methods/techniques, see Supplementary Table 1. We did not fit data from ATP-saturated RBCs because equilibrium theory is not suitable to model active ATP-driven membrane fluctuations [17]."

7. MIET/GIET is essentially a single-point measurement technique. While the approach promises high temporal and axial motion sensitivity, correlation with other lateral locations wouldn't be possible. How does this limitation compare with optical techniques such as diffraction phase microscopy (DPM), which can easily make wide-field measurements at $> 100 \text{ Hz}$?

In principle, MIET/GIET has the capacity to also measure spatio-temporal correlations. There are two possible ways to achieve this: the first one is based on fast scanning confocal microscopy which easily achieves line scan rates of several kHz. This allows for observing spatial correlations along a line with ca. millisecond temporal resolution. In the revised manuscript, we added one scanning measurement on a GUV to demonstrate this point the capability of MIET/GIET to measure also spatial correlations. In this measurement, we scanned a small square area with a frame rate of $24 \text{ ms}/\text{frame}$ ($\sim 40 \text{ Hz}$) and determined spatio-temporal correlations of membrane fluctuations between different lateral positions

(see Supplementary Figure 12). The second way is to use wide-field fluorescence lifetime imaging cameras (Oleksiievets et al., J. Phys. Chem. A, 2020, 124, 17, 3494–3500), which were, until recently, not readily available with a sensitivity as required by FCS. However, new developments of SPAD array technology promise to make spatially resolved and fast MIET/GIET measurements much more feasible. The only limitations of MIET/GIET are that it is fluorescence-based, and that it has a limited dynamic range of height measurements (between 5 and 150 nm).

8. On page 11 of the manuscript, it is mentioned that the dynamic range of MIET/GIET is 160/20 nm. This makes this technique suitable only for basal surfaces. Literature <https://opg.optica.org/ol/abstract.cfm?URI=ol-39-20-6062> indicates that top and bottom surfaces of RBC display different thermal fluctuations due to close contact with the surface on which the cell is resting. The very nature of this approach will measure only subdued membrane fluctuations in intact biological cells. Please comment.

We agree with the reviewer that the bottom surface will be affected by the substrate, which will dampen membrane fluctuations. Our measured membrane fluctuation amplitudes (11 nm for ATP+ RBCs and 7 nm for ATP- RBCs) are much smaller than values obtained with other methods that measure them on the apical side of an RBC. For example, Monzel et al. (Nature Commun. 2015, 6, 8162) obtained values of 74 nm (for ATP+) and 41 nm (ATP-) using DODS, while Park et al. (PNAS, 107, 1289-1294) obtained values of 48 nm (ATP+) and 32 nm (ATP-) using DPM. We have added a corresponding passage and the citations to the RBC measurement section:

“The measured fluctuation amplitudes for both ATP-saturated and ATP-depleted cells as obtained with MIET-FCS are significantly smaller than those obtained using other methods, such as DODS (74 nm for ATP+ and 41 nm for ATP-) [21] or DPM (48 nm for ATP+ and 32 nm for ATP-) [3]. The reason for these smaller amplitudes in our measurements is due to the fact that we can measure only the basal membrane of the RBCs, which is influenced by the presence of the substrate that dampens these fluctuations [53].”

9. How can one use this approach to quantify fluctuations of top or sub-cellular surfaces such as nuclear membrane in intact cells.

MIET/GIET is a near-field technique, which restricts its applicability to samples situated in close proximity to the surface. Thus, it is not suitable for measuring apical membrane fluctuations. For the nuclear membrane, there are cell types for which the basal nuclear envelopes come close enough to the surface so that MIET measurements become possible, see Chizhik et al., ACS Nano, 2017, 11, 12, 11839–11846.

10. On page 15, the statement “This technique holds great promise for assessing cellular functioning in response to drugs and therapies, providing insights into health, mechanical properties, and treatment responses for improved healthcare outcomes. Our method has vast potential for applications in the study of biological membranes both in vitro and in vivo.” is a very general/broad statement. The authors should be more specific and make more realistic claims. For example, in vivo studies with a dynamic range of 160/20 nm (MIET/GIET) is a bit of a stretch.

We agree with the referee, that this statement was too broad and unpecific. We have changed this paragraph to: “We hope that MIET/GIET-FCS will become a powerful and versatile tool for studying membrane biophysics, in particular nanoscale structure and microsecond fluctuations, in a broad range of contexts and applications.”

REVIEWERS' COMMENTS

Reviewer #1 (Remarks to the Author):

The revisions address all my points with clarity. I congratulate the authors to this intriguing new application of MIET.

Reviewer #2 (Remarks to the Author):

I thank the Authors for addressing the raised issues. In the now included inset to Fig 2d, the ACF from lateral diffusion is still visible, but the amplitude is expectedly tiny. To help the readers with less of FCS background, I suggest the following last minor amendments:

- Mention that the amplitude scales with $1/N$ (N being # of molecules in the focal volume), or even faster due to uncorrelated noise, citing relevant sources.
- (p.7) “the resulting auto-correlation function(ACF) measured in FCS experiments is [solely determined] - > [dominated] by the vertical position fluctuations of the membrane”

After that I am happy with the manuscript being published.

Iztok Urbancic

Reviewer #3 (Remarks to the Author):

The authors have adequately attended to all the comments provided by the reviewer, incorporating suitable modifications into the revised manuscript. The current form of the manuscript is deemed suitable for publication.

Reviewer #1 (Remarks to the Author):

The revisions address all my points with clarity. I congratulate the authors on this intriguing new application of MIET.

We thank the reviewer for the positive assessment.

Reviewer #2 (Remarks to the Author):

I thank the Authors for addressing the raised issues. In the now included inset to Fig 2d, the ACF from lateral diffusion is still visible, but the amplitude is expectedly tiny. To help the readers with less of FCS background, I suggest the following last minor amendments:

- Mention that the amplitude scales with $1/N$ (N being # of molecules in the focal volume), or even faster due to uncorrelated noise, citing relevant sources.

We have mentioned this and included citation as requested.

- (p.7) "the resulting auto-correlation function (ACF) measured in FCS experiments is [solely determined] -> [dominated] by the vertical position fluctuations of the membrane"

We have changed the expressions as requested.

After that I am happy with the manuscript being published.
Iztok Urbancic

We thank the reviewer for the positive assessment.

Reviewer #3 (Remarks to the Author):

The authors have adequately attended to all the comments provided by the reviewer, incorporating suitable modifications into the revised manuscript. The current form of the manuscript is deemed suitable for publication.

We thank the reviewer for the positive assessment.